# Extreme value theory as a framework for understanding mutation frequency distribution in cancer genomes

**Natsuki Tokutomi**[1]*, **Kenta Nakai**[1,2], **Sumio Sugano**[3,4]

**1** Department of Computational Biology and Medical Science, Graduate School of Frontier Science, University of Tokyo, Kashiwa, Chiba, Japan, **2** Human Genome Center, Institute of Medical Science, University of Tokyo, Minato-ku, Tokyo, Japan, **3** Medical Research Institute, Tokyo Medical and Dental University, Bunkyou-ku, Tokyo, Japan, **4** Future Medicine Education and Research Organization, Chiba University, Chiba, Chiba, Japan

* evt.cancer.genome@gmail.com

**Data Availability Statement:** Data used in this work are available from the following links. The reformatted files that can be directly processed by our scripts are provided as supplemental files. For details, see S1 Appendix. ICGC: https://icgc.org

## Abstract

Currently, the population dynamics of preclonal cancer cells before clonal expansion of tumors has not been sufficiently addressed thus far. By focusing on preclonal cancer cell population as a Darwinian evolutionary system, we formulated and analyzed the observed mutation frequency among tumors (MFaT) as a proxy for the hypothesized sequence read frequency and beneficial fitness effect of a cancer driver mutation. Analogous to intestinal crypts, we assumed that sample donor patients are separate culture tanks where proliferating cells follow certain population dynamics described by extreme value theory (EVT). To validate this, we analyzed three large-scale cancer genome datasets, each harboring > 10000 tumor samples and in total involving > 177898 observed mutation sites. We clarified the necessary premises for the application of EVT in the strong selection and weak mutation (SSWM) regime in relation to cancer genome sequences at scale. We also confirmed that the stochastic distribution of MFaT is likely of the Fréchet type, which challenges the well-known Gumbel hypothesis of beneficial fitness effects. Based on statistical data analysis, we demonstrated the potential of EVT as a population genetics framework to understand and explain the stochastic behavior of driver-mutation frequency in cancer genomes as well as its applicability in real cancer genome sequence data.

## Introduction

### The "Big Bang" model of cancer development and population genetics of cancer cells

To deconvolve complex biology of cancer, it is useful to trace the temporal order of population dynamics events of cancer cells as well as the underlying somatic genetic events [1]. In cancer, a long mutation and selection process precedes a rapid population increase that results in a clonal expansion which will be observed as formation of a tumor. This model is called the "Big

COSMIC: https://cancer.sanger.ac.uk/cosmic
CHANG: https://github.com/taylor-lab/hotspots/
blob/master/LINK_TO_MUTATIONAL_DATA
RTCGA: https://rtcga.github.io/RTCGA/ IntOGen:
https://www.intogen.org/search DoCM: http://
docm.info.

**Funding:** SS received funding (grant number:
17934018) from the National Bioscience Database
Center (NBDC) of the Japan Science and
Technology Agency (JST)(URL: https://
biosciencedbc.jp/en/). The funder had no role in
study design, data collection and analysis, decision
to publish, or preparation of the manuscript.

**Competing interests:** The authors have declared
that no competing interests exist.

Bang" model of cancer development (Fig 1A) and first proposed for the colorectal cancer growth [2].

Unlike population genetics theories of familial human cancers [3], the population genetics of cancer cells focuses solely on dividing somatic cells within a human individual. Those cells reproduce themselves by asexual somatic cell division and thus are not Mendelian population [4]. However, population genetics models such as SSWM [5], Moran process [6], and branching process [7] are powerful tools for describing population dynamics of cancer cells.

Mutation and selection are two driving forces of cancer evolution. The mutation-selection balance (MSB) model [8–11] assumes infinite population, presence of epistasis [12] and dominance effect. Implication of this model to cancer evolution is enormous [13, 14]. In this model, the loss of genetic variation by selection is equal to the gain of it by mutation. The increase in fitness (i.e., fitness effect) that a mutation confers to the organism is defined as a selection coefficient which may be affected by several types of epistasis [8–10, 15]. In addition, the dominance effect in diploid organisms such as human cancer cells is quantified as dominance coefficient (Fig 1C) [8, 9, 15–17].

According to the "Big Bang" model, the "Big Bang" event separates the cancer development process into two parts: the clonal expansion phase and the preclonal evolution phase. In the clonal expansion phase, multiple tumor subclones are formed with a single expansion of cell population at an early stage of tumor growth (Fig 1A) [2]. In this model, these cells not only acquire clonal lesions that are shared among many cells within the tumor but also acquire subclonal lesions that are observed only in a fraction of the tumor cells at almost the same time. These subclonal lesions are not subjected to stringent selection in cancer evolution (i.e., neutral evolution), thus leading to a state of intra-tumor-heterogeneity (ITH). However, it is known that cancer as a whole is a Darwinian evolutionary system, in which driver mutations undergo selective evolution and passenger mutations undergo neutral evolution [18]. Population genetics studies on cancer evolution in this period have been performed elsewhere, typically with emphasis on neutral evolution, ITH and an increasing population size [7, 19–21]. In contrast, in the tumor initiation step, the preclonal evolution of cancer cells before the population expansion, where the rate of such increase is so small that it is unobservable, has not been sufficiently addressed thus far.

## Preclonal evolution of cancer cells precedes the "Big Bang"

Preclonal evolution before the expansion of the cell population is thought to be selective, with cancer cells acquiring driver mutations that extremely increase cellular fitness [22]. All cancer driver mutations are beneficial for the survival of cancer cells, and acquisition of these mutations brings fitness gain to cancer cells [23]. Among population genetics theories that are able to describe such behavior of the fitness effects of beneficial mutations, the SSWM foundation provides a first-approximation fitness landscape of beneficial mutations [24, 25]. As a specialization of the MSB model for asexual population such as human cancer cells [26, 27], this model assumes that beneficial mutations are introduced and fixed in the population successionally [11, 28, 29] and have independent fitness effects: no epistasis [30] and no dominance (i.e., epistasis and dominance effects are "additive")(Fig 1C, S1B Fig) [31]. In this framework, all beneficial mutations that occur in the population will increase fitness monotonically: i.e., they are on "selectively accessible" paths [25]. Thus, their adaptive walk is typically short.

The SSWM foundation is mainly characterized by two key assumptions: strong selection and weak mutation (Fig 1D) [5]. Here, "strong selection" is equivalent to excluding neutral mutations from consideration. Because all cancer driver mutations are beneficial, the *strong selection assumption* holds for these mutations. In contrast, "weak mutation" means excluding

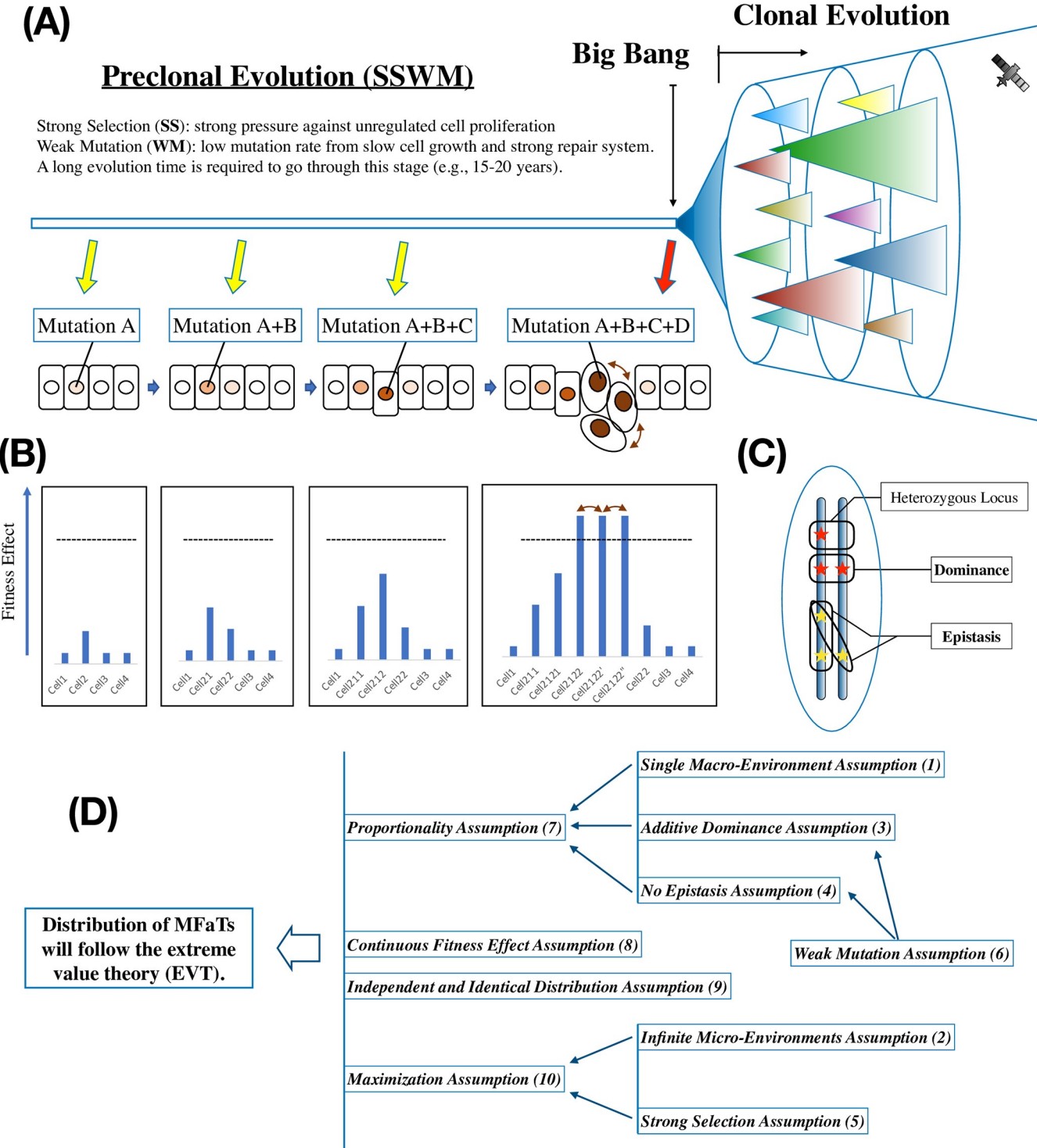

**Fig 1. Extreme value theory is applied to modeling the fitness landscape of preclonal cancer cells.** (A) Overview of cancer evolution. According to the Big Bang model of colorectal tumor growth, a tumor originates from a single clonal expansion. Clonal evolution proceeds after a Big Bang that produces a single clone and multiple subclones. The majority of the subclones are generated at the Big Bang, and only a fraction of them are generated later. These subclones do not experience stringent selection. The period before the Big Bang, on the other hand, is called preclonal evolution. We assume a small and constant population size on cancer precursor cells at this stage. We model population dynamics at this stage with the SSWM regime. The transition from the preclonal dynamics to the Big Bang is modeled with greedy adaptation. The triangles in the siphon indicate subclones in the tumor cell population. The rectangles and circles indicate normal and malignant cells. In the schematic, mutation *A*, for example, is at position *i* in gene *X*. Similarly, mutation *B* can be either at position *j* in gene *Y*, or at position *i'* in gene *X*. (B) In the schematic, the dotted lines indicate fitness thresholds that allow cells to cause a clonal expansion in the given

environment. We assume that the cell which had reached the threshold and started the clonal expansion has the highest fitness level among the preclonal mutant cells. (C) Non-additive mutation effects in diploid genetics. Fitness effects of mutant alleles at the same locus and at different genomic loci can interact resulting in non-additive mutation fitness effects. A heterozygous mutant locus contains a mutant allele and a wildtype allele. A homozygous mutant locus contains two mutant alleles whose fitness effects can interact. In the mutation-selection balance (MSB) model, such fitness effect interaction is called dominance effect. In contrast, fitness effect interaction at any different genomic loci is called epistasis. We distinguish dominance effect from epistasis. Stars indicate mutations. The circle indicates a cell. The vertical bars indicate a copy of the cell's genome. (D) Dependence relations among the ten assumptions. In the schematic, $A \rightarrow B$ means $A$ is a prerequisite for $B$. For example, the *Additive Dominance Assumption* is necessary for the *Proportionality Assumption*. Only a subset of all dependence relationships is shown.

cases in which multiple mutations exist simultaneously within the population. In this assumption, we consider only a genotype that is made by introducing a single mutation to a single wild-type genome. We model population adaptation as a repetition of this step, in which a genotype that is selected from such genotypes becomes a new wild type in the population. Although this assumption always holds true for a microscopic landscape with a small population size, it will be violated if multiple clones interact and multiple mutations compete against each other during their fixation process (clonal interference) [32]. In the preclonal evolution of cancer cells, the *weak mutation assumption* holds because we assume a small, genetically homogenous ancestral cell population of the tumor sample of interest.

An additional but fundamental component of the SSWM foundation is the *additive dominance assumption* (Fig 1D). Here we refer to dominance as non-additive effect of mutations at a single locus in a diploid genome, separating it from epistasis [33] that involves multiple genomic loci such as different genes on the same pathway. Human cancer cells are often assimilated to a population of asexual dividing diploid organisms and thus can have dominance effect at any genomic loci (Fig 1C). In the MSB model, the degree of the non-additivity in this context is quantified by dominance coefficient that describes fitness effect of the heterozygous mutant allele on an additive scale (S1B Fig) [34]. In this study, the dominance of a beneficial mutation is assumed additive as long as its fitness effect is small [31, 35, 36].

In the preclonal evolution of cancer, a non-silent mutation within a driver gene can have small fitness effect due to the complexity of cancer pathology. Even within a single cancer type, tumor micro-environments of each tumor sample are sometimes greatly different from each other [37, 38]. As a result, it is possible that a driver mutation which has caused a Big Bang in one tumor sample remains to have only limited fitness effect in another. In addition, elements of tumor micro-environments including resource availability may differ depending on location and change over time [39–43]. This will partly explain the genetic diversity of cancer including difference in necessary driver genes among tumors. And fundamentally, a clonal expansion of the cell population which is the direct cause of the Big Bang is expected to occur only once in the Big Bang model [2]. This means that the fitness effects of beneficial mutations induced before the Big Bang are relatively small in comparison with the driver mutation which is the direct cause of the single clonal expansion. Therefore, we assume that the dominance effect of beneficial mutations including those in driver genes in a preclonal cancer cell population is small and to remain approximately additive before the Big Bang (the *additive dominance assumption*).

In contrast, driver mutations which are causative of the Big Bang are often characterized by non-additive fitness effects in the event. These "causative" driver mutations are each a direct cause of the Big Bang in the tumor sample of interest, and thus likely to have non-additive interaction such as dominance and recessiveness (S1A and S1B Fig). Of the two categories of cancer driver genes (i.e., oncogenes (OGs) and tumor suppressor genes (TSGs)) [44], the former such as *NRAS* and *KRAS* is likely to have dominant driver mutations [45], while the latter such as *Rb* and *TP53* is likely to have recessive driver mutations [46–48]. These events

represent not only breaking of the *additive dominance assumption*, but also breaking of the *weak mutation assumption* since they cause a rapid population expansion. Similarly, any such expansion-causing oncogenic events including chromosomal missegregation [49–52], epigenetic lesion [53], copy number alteration [54], mutant driver homozygosity [55], and loss-of-heterozygosity [48, 56, 57] are all assumed to break the *weak mutation assumption*.

## Breaking of SSWM dynamics primes clonal expansion of cancer cell population

An expansion of the cancer cell population size generates multiple clones with multiple different beneficial mutations, thus causing clonal evolution [20, 58] of the newer cells. Here, the *weak mutation assumption* is violated and the population dynamics of the cells is thought to shift from SSWM to greedy adaptation (Fig 1A) [59, 60]. A phenomenon wherein mutations with greater fitness effects are certainly fixed in the population, resulting in increased repeatability of the fitness trajectory of cells in such a situation [61], has been confirmed by an *Escherischia coli* experiment [62]. The fact that a limited number of driver mutations has been observed repeatedly in multiple cancer samples suggests not only the repeatability of the oncogenic process, but also the underlying evolutionary structure itself.

An increase in population size not only strengthens the deterministic traits [61] of the population by clonal interference but also enables its fitness valley crossing [25, 63]. This gives rise to an escape genotype that is not on the selectively accessible paths in SSWM, thereby counteracting the deterministic traits of the population. Stochastic tunneling, a critical element in the valley-crossing mechanism, enables fixation of deleterious mutations as well as neutral evolution [20, 64]. These theoretical aspects of cell population behavior have been validated by *E. coli* experiments [65]. In cancer, such population dynamics corresponds to the subclonal evolution of passenger mutations that enhances ITH.

Importantly, an increase in population size also introduces another complexity in population dynamics of cancer cells. Since the *weak mutation assumption* is violated, a newer mutation can arise when an older mutation has not been fixed. With multiple mutations in a single cell present, the effects of those mutations can interact resulting in a non-additive fitness effect at different loci (i.e., epistasis) [33, 66]. The effect of epistasis on fitness landscape on a practical time scale has been analytically and computationally examined in relation to the weak-mutation limit and the distribution of fitness effects [67]. Roles of epistasis in cancer initiation are highlighted elsewhere [68, 69].

Epistatic effect introduced by violation of the *weak mutation assumption* changes fitness landscape of each cell in the population. This may result in a rugged fitness landscape [33, 70, 71] in which previously "beneficial" driver mutations become no longer beneficial. For example, in a tumor with an already activated driver pathway, an activating mutation in a driver gene on that pathway no longer contributes to tumorigenesis. This corresponds to violation of the *strong selection assumption* that contributes to the neutral evolution [20] of cancer driver genes. This in combination with the increasing population size reinforces the basis of applicability of the Moran process [6, 20] and, if the increase is exponential, the branching process [7, 33, 44, 47, 68, 72]. Neutral evolution itself enhances ITH [19].

## Extreme value theory as a framework for explaining mutant sample frequency distributions of cancer driver mutations

Among the evolutionary models mentioned above, this manuscript focuses on SSWM (and the ten assumptions: see Discussion and Fig 1D) to explain the acquisition of driver mutations by cancer cells. According to the mutational landscape by Gillespie, the fitness of a wild type

allele at a given gene locus tends to lie on the right side of a distribution because this fitness is usually high, and the fitness of beneficial mutations lie further to the right [5, 31]. In this setting, the fitness values are extreme, and thus their statistical behavior is described by EVT [73, 74].

The first application of EVT to cancer preclonal evolution examined the proliferation of stem cells in intestinal crypts [75]. Within the internal space of an individual crypt, the adaptation and evolution of the cell population will be completely independent of the states of different crypts. In our analysis, we employ an analogy that compares an intestinal crypt and a tumor sample. Similar to a single crypt in the colon, a tumor sample has multiple cells that are each independent evolutionary players.

In cancer genomics, "mutation frequency" in general refers to the variant allele frequency (VAF) which is defined as a fraction of mutant reads to total reads at a given genomic site.

$$\text{VAF} = \#\{\text{mutated reads}\} \ / \ \#\{\text{total reads}\} \qquad (1)$$

This is approximately proportional to the mutant cells contained in a single sample if the tumor sample purity is assumed constant. Recent studies have shown that the neutral evolution of the cancer genome results in a power law distribution of tumor bulk-sample VAFs reported by a next-generation sequencer [76, 77].

In contrast, we define MFaT as a normalized frequency of mutant samples within a single dataset.

$$\text{MFaT} = \#\{\text{mutated tumors}\} \ / \ \#\{\text{total tumors}\} \qquad (2)$$

This measure is proportional to patient/sample frequency in the dataset, and thus focuses more on the repeatability instead of the clonality of a mutation. In this study, we adopt MFaT as a proxy for expectation of VAF and fitness effect of a beneficial mutation (the *proportionality assumption*). Since beneficial mutations experience selective evolution in carcinogenesis, the distribution of MFaTs will be different from those of VAFs of neutral mutations. It should be noted that this formulation is valid only with driver mutations in the Big Bang model [2].

## Materials and methods

### Study design

The overall analysis consists of two parts: total tumor analysis and tumor type-specific analysis. The former contains two categories of cancer driver definitions: "driver-gene definition" and "driver-site definition". The "driver-gene definition" refers to a list of gene symbols which are experimentally or computationally identified driver genes. The "driver-site definition" specifies nucleotide positions as well as their gene symbols. In the tumor type-specific analysis, only driver-gene definitions are used. An intersection set is calculated for every combination of a large-scale cancer genomic dataset and a cancer driver definition. Taking intersection of a given dataset with a "driver-gene definition" equals to applying filtering by gene symbols (symbol-based filtering). On the other hand, taking intersection of a given dataset with a "driver-site definition" equals to applying filtering by genomic positions (position-based filtering).

Throughout the analyses, different mutations in the same gene but at different nucleotide positions are counted separately. For example, if a gene $G$ contained only non-silent mutations at positions $P$ and $Q$, the gene $G$ has two mutations. Sample frequencies in each dataset are calculated based on combinations of $G$ and $P$, distinguishing post-substitution nucleotides (e.g., an A-to-C mutation and an A-to-G mutation at $P$ are distinguished).

## Datasets

In total, we used unique doubleton protein-altering single nucleotide variations (SNVs) that numbered 68 973 from 15 285 patients in the ICGC Release 27 dataset [78], 325 244 from 24 355 tumors in the COSMIC Version 85 dataset [79], and 91 312 from 11 089 samples in the Chang dataset [80]. These groups of unique mutations respectively correspond to groups of unique genomic sites: 68 613 in the ICGC dataset, 322 962 in the COSMIC dataset, and 90 885 in the Chang dataset. These mutations were annotated as being either in the "missense variant," "initiator codon variant," "stop gained," or "stop lost" mutation class. In all datasets, the reference genome version is GRCh37, and the annotation is based on Ensembl v75.

In addition, we used the RTCGA mutations dataset (https://rtcga.github.io/RTCGA/) to enable a tumor-type-specific analysis over cancer clonal mutations identified in an identical sequencing strategy. This dataset includes 178 701 unique somatic protein-altering SNVs from 2 732 samples and corresponding 177 898 unique genomic sites. The samples are collected from eight cancer types: BLCA (127), BRCA (949), HNSC (266), LIHC (191), LUAD (225), PRAD (268), SKCM (340), and THCA (366) (sample sizes indicated in parentheses). The sequencing strategy is whole exome sequencing (WXS) performed with two platforms (Illumina GAIIx, Illumina HiSeq) at six centers (broad.mit.edu, genome.wustl.edu, hgsc.bcm.edu, ucsc.edu, mdanderson.org, and bcgsc.ca). All mutations are annotated as being in either the "missense mutation," "nonsense mutation," "translation start site," or "nonstop mutation" class. We defined mutations with VAFs in the range of [0.25, 1.00] as clonal after a pioneering study in the field [76].

We used the following cancer driver-gene definitions in our analysis: the Mutational Driver definition in the IntOgen Cancer Drivers Database (2014.12) dataset [81], the COSMIC Cancer Gene Census [79], and the Tokheim Oncogenes and Tumor Suppressor Genes [82]. Also, we used the following data as cancer driver mutation site definitions: SNVs whose driver activity is "known" in the IntOgen 2016.5 Driver Mutations Catalog Mutation Analysis dataset [83], SNVs in the DoCM database [84], and amino acid substitution information per gene in a recent study (S4 Table in [85]).

## Data processing

For the RTCGA dataset, mutations were filtered based on the VAF threshold described in the previous section. For the rest of the data, mutations were filtered based on the criteria as in [78–80].

MFaT values for each nucleotide mutation were calculated on the basis of the fraction of mutated sample/patient count in the total samples/patients (for details, see Supplementary Materials in S1 Appendix). In brief, the calculation was performed with special attention to minimize the effect of artificial manipulation to data that would possibly be affecting observed MFaT distributions. To filter out potentially low-confidence mutations, we selected mutations that appeared at least twice in a dataset (total sample/patient counts are based only on sequences containing such mutations). This would have ameliorated the problem of the original datasets which are lacking correction or control of tumor sample purity. Protein-altering mutations were selected based on all possible annotations to avoid unintentionally omitting mutations that are synonymous in one annotation and at the same time protein-altering in another (the *strong selection assumption*). Sample/patient counts were calculated in a manner sensitive to reference/alternate allele combinations (e.g., a T to C mutation and a T to A mutation at the same genomic site are distinguished throughout the analysis).

## The definition and calculation of MFaT

We defined mutant allele frequency among tumors (MFaT) as a frequency of samples within a cancer genome dataset that have a given mutation at a given genomic site (see Eq 2). The MFaT value is defined at any genomic sites corresponding to individual mutations recorded in the dataset. In this formulation, the count of samples having a mutation is normalized by the number of total samples within the dataset, permitting a comparison of mutated sample frequencies between different datasets.

To practically and approximately calculate continuous MFaT values, we utilized multiple total-tumor cancer genome datasets. In general, observed MFaT values will be discrete in any dataset with a sample size less than 2 000. The total-tumor datasets, which typically have thousands of sequencing samples, will provide enough resolution in MFaT values to satisfy the *continuity assumption*. With the aims of ensuring individual mutation observations and reducing computational cost, we restricted our analysis with doubleton (i.e., observed at least twice in each dataset) somatic protein-altering SNVs. A protein-altering mutation in our study is defined as a mutation that changes the protein sequence in either of the annotated transcripts. On mutations that do not satisfy this condition, or non-protein-altering mutations, one can define MFaTs although the *proportionality assumption* does not hold over the values (for details, see Discussion).

To exclude mutations with smaller fitness effects, we restricted our analysis to the mutation sites with MFaT values ranked in the top 200 in the case of total-tumor analysis and top 50 in the case of type-specific analysis. Here, we assigned smaller ranks to greater MFaT values. For tied MFaT observations, we assigned the maximum rank (e.g., for MFaT observations {0.1, 0.3, 0.3, 0.5}, ranks {4, 3, 3, 1} will be assigned). The number of plotted MFaT values after the filtering is shown in the figures as "b," and the effective threshold "th" is shown where relevant.

## The maximization of MFaT

In practice for large-scale cancer genome analysis, it is common to observe multiple mutations on a single genomic site. This is due to the phenomenon that a single site is recurrently mutated to differing bases (substitution) in a set of samples. Since we consider only simple substitutions at this time, three alternate sequences of a given genomic site are possible (e.g., {A, T, C} against a reference base G). This could result in, at maximum, three different MFaT values for one site (e.g., MFaT for G to A, MFaT for G to T, and G to C).

In general, the number of cases of possible cancer cell environments that dominate cancer evolution via selective pressure is unlimited. Thus, the number of possible selective coefficients (i.e., the fitness effect) at a given genomic site is well approximated by positive infinity (the *infinite micro-environments assumption*). This is equivalent to the case where the number of possible alternate sequences of a given genomic site is infinite.

In addition, we assume that the mutational selection coefficient of a given site is maximized among possible alternate values after evolutionary selection (the *maximization assumption*).

$$S_i = \max(S_{i,1}, S_{i,2}, \ldots, S_{i,n}) \qquad (n \to \infty) \tag{3}$$

where *S* denotes a mutational selection coefficient, *i* is the index for genomic sites, and *n* is the number of possible values, respectively. This is consistent with the selectionist idea of the survival of the fittest, and the block maxima model in the extreme value theory.

To suffice for the *maximization assumption* of MFaT observations, we selected the maxima of MFaT values in each genomic site, and excluded the rest from our analysis. As a result, the counts of MFaT values, mutations, and sites will all be equal. Here, we define this process as "maximization" of MFaTs, which will enable more exact and reliable data processing in future

analyses. In the case of a tumor-type-specific analysis, we performed this maximization process for the respective tumor types.

## Parameter estimation

We estimated the generalized extreme value distribution (GEV) and generalized Pareto distribution (GPD) parameters by the maximum likelihood (Nelder-Mead optimization) method using the "evd" R package (i.e., evd::fgev, evd::fpot functions). Initial values for the optimization are set in the GEV parameter estimation in the total-tumor analysis using the ICGC, COSMIC, and Chang datasets (shape: 1.25, location: 0, scale: 0), while they are not set in bootstrap simulations using these datasets. We also used different initial values (shape: 1.0, location: 0, scale: 0) in the total-tumor analysis using the RTCGA dataset.

For parameter estimation of the Pareto distribution, we defined its probability density function (PDF) and maximum likelihood estimators as the following, referring to implementations in the "VGAM" R package. Here, $n$ is the length of the observed data vector.

$$f_{\text{Pareto}}(x) = \frac{\alpha x_m^\alpha}{x^{\alpha+1}} \qquad x > x_m \tag{4}$$

$$\hat{x}_m = \min_i x_i \tag{5}$$

$$\hat{\alpha} = \frac{n}{\sum_{i=1}^n \ln x_i - n \ln \hat{x}_m} \tag{6}$$

## Goodness-of-fit assessment by $\chi^2$ goodness-of-fit test

To assess goodness-of-fit of the GEV and Pareto distributions, we performed a conventional $\chi^2$ goodness-of-fit test over the total-tumor and tumor-type-specific cancer driver MFaTs using the "stats" R package (stats::chisq.test function; rescale.p and simulate.p.value flags were both set to TRUE). The theoretical frequencies of the MFaTs were calculated using parameters estimated by the maximum likelihood method (see Parameter Estimation section) and then compared with the observed MFaT frequencies. In this setting, the null hypothesis $H_0$ stated, "the observed distribution is identical with the theoretical," and the alternative hypothesis $H_1$ stated, "the observed distribution is different from the theoretical." Consequently, a higher significance level such as $p < 0.05$ rejected the $H_0$ and indicated that "the observed distribution is significantly different from the GEV," for example. Conversely, a lower significance level such as $p = 0.50$ weakly supported the notion that the observed and theoretical distributions are alike.

## Bootstrap simulation and confidence interval

Bootstrap simulations over parameter estimators were performed using the "boot" R package (boot::boot function) in the total-tumor analysis. In all analyses, the iteration number is set to $n = 1 \times 10^5$. We did not set initial values in the cases of the ICGC, COSMIC, and Chang datasets, while they were set in the case of the RTCGA dataset (shape: 1.0, location: 0, scale: 0).

We calculated 95% confidence intervals for the GEV shape parameter (i.e., tail index) using simulated total-tumor bootstrap distributions and a variety of calculation methods (in boot:: boot.ci function): the first-order normal approximation (Normal), the basic bootstrap interval (Basic), the studentized bootstrap interval (Student), the bootstrap percentile interval (Percentile), and the adjusted bootstrap percentile (BCa) interval.

## Density plot

For MFaT observations in each total-tumor case, we plotted theoretical probability density functions (PDFs) obtained from the parameter estimation of distributions and empirical PDFs obtained from observed values. We used the "evd" R package (evd::dgev and evd::dgpd functions) to plot GEV and GPD PDFs. Similarly, we defined the PDF of the Pareto distribution by referring to the "VGAM" R package (VGAM::dpareto function) (see Parameter Estimation section). We used the "stats" R package (stats::density function) to draw the empirical density function obtained from observations.

## Data normalization in Q-Q Plot

We calculated a theoretical value of an observed MFaT value $x_k$ with $Q(k/(N + d))$, where $Q(\cdot)$ is the quantile function of GEV, $k$ is the rank of the observation when the values are ranked from the smallest, $N$ is the number of observations ($N = b$ in plots), and d is the numerator correction factor (both $b$ and $d$ are shown in plots). The quantile function is defined by an inverse function of the cumulative distribution function (CDF) of GEV. We used the "evd" R package (evd::qgev function) to calculate the quantile function values.

## Formulation of Fréchet plot

Based on the Weibull plot introduced in [86], we developed and drew a Fréchet plot (see demonstration in Supplementary Materials in S1 Appendix). After the assessment of the goodness-of-fit of the GEV distribution to the observations, we calculated slope and intercept values in a linear regression. In the cases of tumor-type-specific and gene-specific Fréchet plots, we added an additional data point MFaT = 1.0 to the observations in calculating each value of the empirical distribution function $F_{emp}(x)$ against the observed MFaT $x$. This satisfied the condition $0 < F_{emp}(x) < 1$ so that the $y$-axis value $y(x)= -\ln(-\ln F_{emp}(x))$ is finite. In the gene-specific cases, we plotted only genes with more than two unique MFaT values. In such cases, we analyzed skin cutaneous melanoma (SKCM) tumors separately that are more enriched in mutations.

## Bayesian parameter estimation by Markov Chain Monte Carlo simulation

In Bayesian extreme value analysis, we estimated tumor-type-specific GEV parameters using the Markov Chain Monte Carlo (MCMC) approach. We used the "evdbayes" R implementation for this purpose. Prior distributions of respective GEV parameters (i.e., shape, scale, and location) were assumed independent and normal. This means that the variance-covariance matrix that is used to calculate prior distribution is diagonal. Also, this distribution is equal to the trivariate (i.e., with three variables) normal distribution with variables that are mutually independent. In addition, we determined the prior parameters regarding the results of the bootstrap simulation, which are dependent on observed data. This means that the prior distribution used in this step is "informative."

Specifically, the normal parameters in the informative priors were set as follows (mean ± standard deviation): shape $1.0 \pm 0.5$, scale $2.5 \times 10^{-4} \pm 3.2 \times 10^{-4}$, and location $1.5 \times 10^{-3} \pm 1 \times 10^{-4}$. The standard deviations of proposal distribution in MCMC were set as follows: shape 0.01, scale $1 \times 10^{-5}$, and location $1 \times 10^{-4}$. The burn-in number of MCMC was set to 5 000, whereas the number of iterations was $1 \times 10^5$. The initial values of the MCMC simulation were set equal to the expectation of the normal priors. However, for scale parameters in tumor-type-specific parameter estimation, we set the next value $\tau$ as the initial values considering the loss of genetic diversity in sequencing samples due to the shrinking sample

size, which has been a problem in tumor-type-specific analyses.

$$\tau = 1/[\text{Sample size of the tumor type}] \qquad (7)$$

After the MCMC simulation, we estimated tumor-type-specific GEV parameters using an expected à posteriori (EAP) estimator.

## Estimation of mutational fitness effects by MFaT

Consider the relative fitness $W_{i,A}$ of a cancer cell with a post-mutational sequence $A$ at a certain genomic site $i$. We define this value using the classic evolutionary selection coefficient, or fitness effect, $S_{i,A}$:

$$W_{i,A} = 1 + S_{i,A} \qquad (8)$$

With this definition, we formulate the selection coefficient whereby the cell's genome is mutated from a pre-mutation (or reference) sequence $R$ (with length 1 bp) to a post-mutational sequence $A$ [27].

$$S_{i,R \to A} = \frac{W_{i,A} - W_{i,R}}{W_{i,R}} = \frac{S_{i,A} - S_{i,R}}{1 + S_{i,R}} \qquad (9)$$

Then, we consider cancer cell environments that determine selective pressure throughout cancer evolution. We consider the set of all possible environmental states $\Theta$ and its elements $\theta$ that determine the value of a selection coefficient of a cancer cell with a given genotype. Next, we define the selection coefficient whereby the cell mutates from a pre-mutational sequence $R$ to a post-mutational sequence $A$ at a genomic site $i$ within a given environmental state $\theta$:

$$S_{i,R \to A,\theta} = \frac{S_{i,A,\theta} - S_{i,R,\theta}}{1 + S_{i,R,\theta}} \qquad (10)$$

In addition, we assume that the fitness effect of a cancer driver mutation after preclonal evolution is maximized (the *maximization assumption*). In other words, the fitness effect of a given mutation is the maximum of many alternative values that are possible depending on the combinations of alternate sequence information and possible cancer cell environments. Under this formulation, the mutational selection coefficient (MSC), whose values are unique to cancer driver mutation sites, is defined by:

$$S_i = \max(\{S_{i,R \to d,\theta}\}_{d \in D, \ \theta \in \Theta}) \qquad (11)$$

Here, $D$ is the set of post-mutational DNA sequences that are possible at the site $i$, and $d$ is its single element. $D$ is dependent on the genomic site $i$ and the scope of consideration of mutational classes. For example, in the case considering only simple substitutions, then $D = \{A, T, C\}$ if the pre-mutational sequence $R$ was $G$. In this case, the size (i.e., the number of elements) of $D$ is $\#\{D\} = 3$. In contrast, the set $\Theta$ is independent of genomic sites and pre- and post-mutational sequences. Here, we assume the size of the set $\#\{\Theta\}$ is infinite (i.e., the number of possible micro-environmental states is numerous) (the *infinite micro-environments assumption*). Then, the number of possible selection coefficients $n = \#\{D\} \cdot \#\{\Theta\}$ for a given site $i$ tends to infinity, and $n \to \infty$ holds.

Let the individual values of $S_{i,R \to d,\theta}$ at respective genomic sites $i$ be continuous and independent and identically distributed (IID) about $D$ and $\Theta$ (the *IID assumption* and the *continuity assumption*). Based on the maximization, IID, and *continuity assumptions*, the block maxima model in extreme value theory is valid over $S_i$, demonstrating that $S_i$ converges to the

generalized extreme value distribution (GEV) as $n \to \infty$. Finally, as this limit holds due to the *infinite micro-environments assumption*, the probability distribution of $S_i$ will be GEV over the set of genomic sites $J(i \in J)$.

Here, we assume that MSC at a site $i$ is proportional to the MFaT of the site $i$ (the *proportionality assumption*).

$$S_i \propto [\text{MFaT}]_i \tag{12}$$

We generalize this by considering normalizing constants of MFaTs, $\sigma_0$ and $\mu_0$.

$$S_i = \frac{[\text{MFaT}]_i - \mu_0}{\sigma_0} \tag{13}$$

In general, fitness $W$ is unobservable, and so is the relative fitness $W_{i,A}$ by a mutation at site $i$ in preclonal cancer cells. Thus, here we heuristically set $\mu_0 = 0$ and $\sigma_0 = 1$. Then, the estimator of MSC at site $i$ is given by

$$\hat{S}_i = [\text{MFaT}]_i \tag{14}$$

## EAP estimation of MSCs by GEV-binomial model

We consider the probability distribution of the sample count $k$ of tumor samples which have a certain post-mutational sequence at a given genomic site $i$. Since $0 < \text{MFaT} < 1$, here we consider the binomial distribution $\text{Binom}(m, S_i)$ with $S_i$ (the MSC at site $i$) as a ratio parameter and $m$ (the number of tumor samples) as a size parameter. We update the Bayesian knowledge of the parameter $S_i$ in our Bayesian framework using observations of $k$ and $m$. Specifically, we assume GEV as a prior distribution over $S_i$, and then we calculate the posterior distribution using discrete observations at individual genomic sites.

The prior distribution of $S_i$ is expressed using the PDF of GEV $f_{\text{GEV}}(s)$ with three parameters (shape $\xi$, scale $\sigma$, and location $\mu$) as follows:

$$f_{\text{GEV}}(s) = \frac{1}{\sigma} A(s)^{-\frac{1}{\xi}-1} \exp\left[-A(s)^{-\frac{1}{\xi}}\right] \tag{15}$$

$$\xi \neq 0 \quad ; \quad A(s) = 1 + \xi \frac{s-\mu}{\sigma} > 0 \tag{16}$$

Here, $A(s)$ is a function of $s$, the argument of the PDF. The likelihood at each value of tumor sample counts, $k = 0, 1, 2, \ldots, m$, is expressed using the probability mass function (PMF) of the binomial distribution $f_{\text{Binom}}(k)$ as follows:

$$P(X = k | S_i = s) = f_{\text{Binom}}(k) = \binom{m}{k} s^k (1-s)^{m-k} \tag{17}$$

From Bayes theorem and appropriate assumptions over PMF and PDF, we assume the following equation over the posterior distribution of $S_i$

$$P(S_i = s | X = k) = \frac{P(X = k | S_i = s) P(S_i = s)}{P(X = k)} \tag{18}$$

Here, the necessary assumptions are

$$P(S_i = s) = f_{\mathrm{GEV}}(s) \tag{19}$$

$$P(X = k) = \sum_i P(X = k|S_i) \tag{20}$$

For the stepwise calculation of $P(X = k)$, we need to consider tumor sample counts at all genomics sites. However, this is easily achieved by normalization of the numerator $P(X = k|S_i = s)P(S_i = s)$. Finally, we estimate $S_i$ by the expected à posteriori (EAP) estimation and obtain its 95% confidence interval (EAP ± 1.96 APSD) by calculating the à posteriori standard deviation (APSD).

## Results

### The probability distribution of driver MFaT is likely a Type II extreme value distribution

A visual inspection of the distribution shape using a density plot (Fig 2A and S2A Fig) suggested that the probability distribution of cancer driver mutation MFaTs is approximately equal to the extreme value distribution (for details, see Materials and methods). A Q-Q plot with Sugano normalization (see Materials and methods) confirmed that the driver MFaT distribution is approximately described by the extreme value distribution given the relationship between observed and theoretical values (Fig 2B, S2B Fig). In addition, the arrangements of data points in Fréchet plots are approximately linear ($R^2 > 0.92$ in symbol-based filtering cases and $R^2 > 0.97$ in position-based filtering cases), suggesting driver MFaTs follow the Fréchet distribution (Fig 2E, S2E Fig). The results of a bootstrap simulation (Fig 2C and 2D, S2C and S2D Fig) over the tail index as the shape parameter of the generalized extreme value distribution (GEV) revealed that the tail index of driver MFaTs is common among different cancer genome datasets (1.25 ± 0.62 with a 95% confidence interval), and its value is positive (Type II extreme value distribution or Fréchet distribution). Also, the result suggested that the tail index can contribute to the estimate of systematic biases within a dataset, and in this case, the ICGC dataset harbors greater noise compared to other datasets (Fig 2C, S2C Fig).

Finally, the results of a $\chi^2$ goodness-of-fit test did not exclude the possibility that the actual distribution of driver MFaTs is GEV (with a significance level $\alpha = 0.05$) (Tables 1 and 2). In this statistical test, the null hypothesis $H_0$ that may be rejected is "the observed distribution is identical with the theoretical," and the alternative hypothesis $H_1$ that may be accepted is "the observed distribution is different from the theoretical." Thus, the former null hypothesis ($H_0$) is our claim. For example, in the case of ICGC-DoCM in the driver-site analyses, we cannot conclude that the two distributions are different because the null hypothesis is not rejected at a significance level $\alpha = 0.05$ ($p = 0.51 > 0.05$). This does not state that the two distributions are identical but weakly supports the claim that the distribution of driver MFaT is GEV.

Collectively, the total-tumor analysis, using both symbol-based and position-based filtering methods, strongly suggested that cancer driver mutation MFaTs in total-tumor datasets are appropriately modeled by the Type II extreme value (Fréchet) distribution.

### Bayesian MCMC approach confirms that tumor-type-specific driver MFaT distributions are also Fréchet type

We then asked whether driver MFaT distributions were also of the Fréchet type in each tumor type. We used the RTCGA dataset for this analysis regarding data availability. The results of

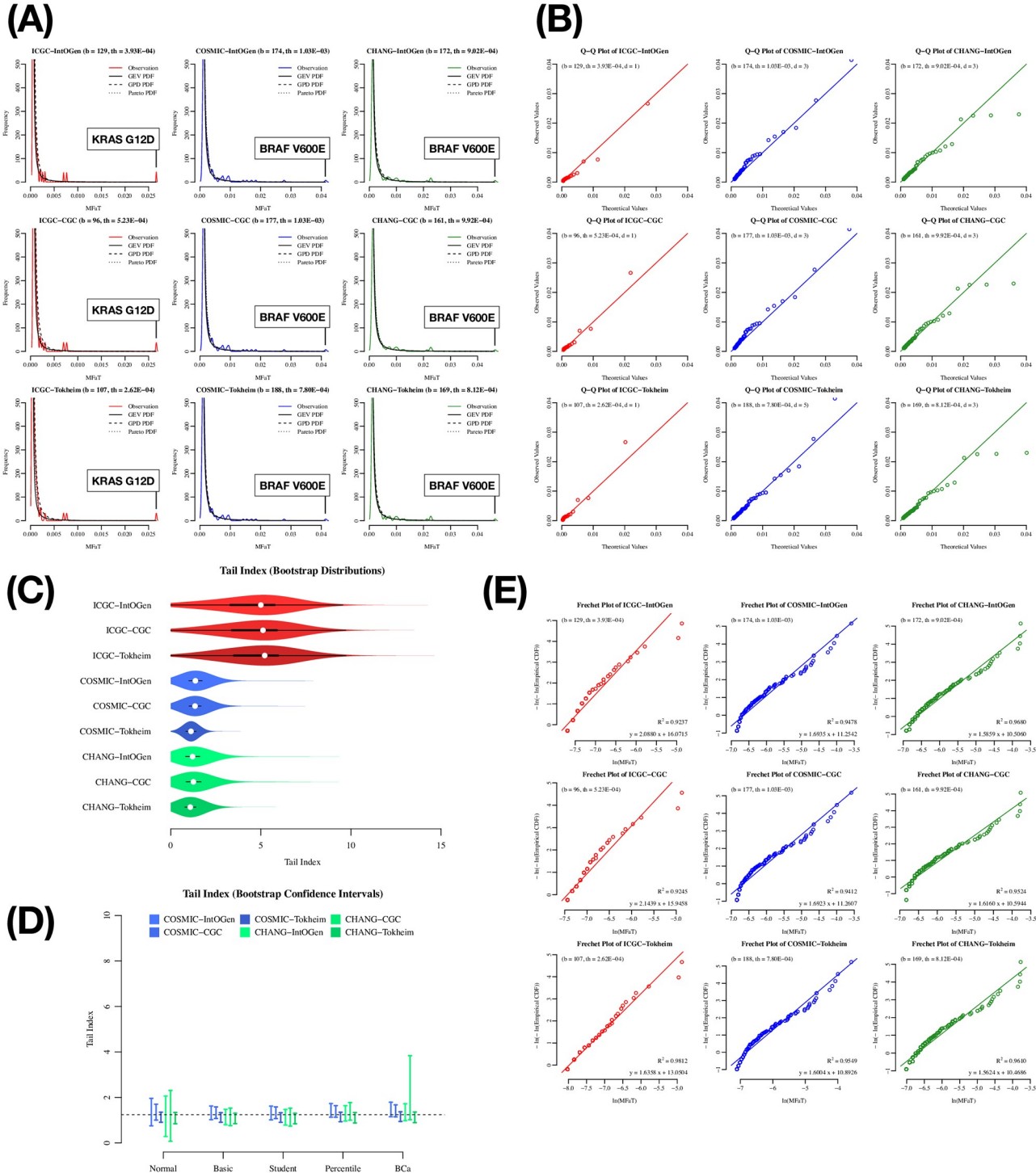

**Fig 2. Exploratory plots on cancer driver mutation MFaTs in the total-tumor analysis with symbol-based filtering.** In these figures, the "ICGC-IntOgen," for example, denotes a set of mutations as an intersection of the ICGC mutations with the IntOgen driver-gene definition (i.e., symbol-based filtering)(for details, see Materials and methods). (A) Density plot of MFaTs. The colored solid line is probability density of observations, the black solid line is the probability density function (PDF) of GEV, and the dotted lines are the PDFs of the GPD and Pareto distributions. For each plot, a gene symbol and an amino acid substitution of a mutation with the highest MFaT value is shown in a box. (B) Q-Q plot of MFaTs. Here, "b" denotes the number of genomic sites of beneficial mutations considered, "th" denotes the effective threshold against MFaTs when selecting mutations according to ranks (for details, see Materials and methods), and "d" denotes the parameter in the normalization in the Q-Q plot. The straight line has the equation $y = x$. In the Q-Q plot, each point denotes a pair consisting of an observation and its corresponding theoretical value. (C) Bootstrap

distributions of the tail index. The white point indicates the median of the distribution. The black square shows the first and the third quantiles. (D) Bootstrap confidence intervals. The dotted line shows the value that is in range of all confidence interval cases. (E) Fréchet plot. The $R^2$ values and the equation for the linear regression lines are shown. The symbols "b" and "th" are as defined in (B).

the "total"-tumor analysis using the RTCGA dataset (Fig 3A and 3B) confirmed that estimated values of GEV parameters in the case of the RTCGA dataset are reproducible and independent of differences in filtering methods, as shown with the ICGC, COSMIC and CHANG datasets (Fig 2C and S2C Fig). Tumor-type-specific analyses of eight tumor types (Fig 3C and 3D) confirmed that the results were similar to the results of the total-tumor analysis. Parameter estimation by Bayesian MCMC (Fig 3C) showed the MFaT distributions belonged to the Fréchet type, although some degree of variability in GEV parameters according to the differences in tumor types were observed. In addition, the comparison of histograms of observations to estimated densities (Fig 3D) confirmed that the probability distribution of GEV approximately describes the actual distributions of type-specific driver MFaTs. The arrangements of data

**Table 1. The $\chi^2$ goodness-of-fit test $p$-values of total tumors with *symbol*-based filtering (rank $<$ 200).**

| Combination Case | MFaT | Driver Definition | GEV $p$-value | Pareto $p$-value |
|---|---|---|---|---|
| ICGC-IntOgen | ICGC | IntOgen | 0.280 | 0.005 |
| ICGC-CGC | ICGC | CGC | 0.301 | 0.009 |
| ICGC-Tokheim | ICGC | Tokheim | 0.269 | 0.081 |
| COSMIC-IntOgen | COSMIC | IntOgen | 1.000 | 0.758 |
| COSMIC-CGC | COSMIC | CGC | 1.000 | 0.767 |
| COSMIC-Tokheim | COSMIC | Tokheim | 1.000 | 1.000 |
| CHANG-IntOgen | CHANG | IntOgen | 1.000 | 1.000 |
| CHANG-CGC | CHANG | CGC | 1.000 | 1.000 |
| CHANG-Tokheim | CHANG | Tokheim | 1.000 | 1.000 |

Note that in this setting, a higher significance level such as $p < 0.05$ indicates that the observed and theoretical distributions are different, and a lower significance level such as $p = 0.50$ weakly supports a notion that the two distributions are alike. In this table, the GEV $p$-values are all above 0.25 maintaining the null hypothesis ($H_0$) that the two distributions are identical, while some of the Pareto $p$-values have fallen below 0.05 accepting the alternative hypothesis ($H_1$) that the two distributions are different.

**Table 2. The $\chi^2$ goodness-of-fit test $p$-values of total tumors with *position*-based filtering.**

| Combination Case | MFaT | Driver Definition | GEV $p$-value | Pareto $p$-value |
|---|---|---|---|---|
| ICGC-IntOgen | ICGC | IntOgen | 0.405 | 0.336 |
| ICGC-DoCM | ICGC | DoCM | 0.529 | 0.316 |
| ICGC-Tokheim | ICGC | Bailey | 0.244 | 0.203 |
| COSMIC-IntOgen | COSMIC | IntOgen | 1.000 | 1.000 |
| COSMIC-DoCM | COSMIC | DoCM | 1.000 | 1.000 |
| COSMIC-Bailey | COSMIC | Bailey | 1.000 | 0.672 |
| CHANG-IntOgen | CHANG | IntOgen | 1.000 | 1.000 |
| CHANG-DoCM | CHANG | DoCM | 1.000 | 1.000 |
| CHANG-Bailey | CHANG | Bailey | 0.632 | 0.625 |

Note that in this setting, a lower significance level with higher $p$-value such as $p = 0.50$ weakly supports a notion that the observed and theoretical distributions are alike. In this table, $p$-values are equal or higher in the GEV than the Pareto distribution.

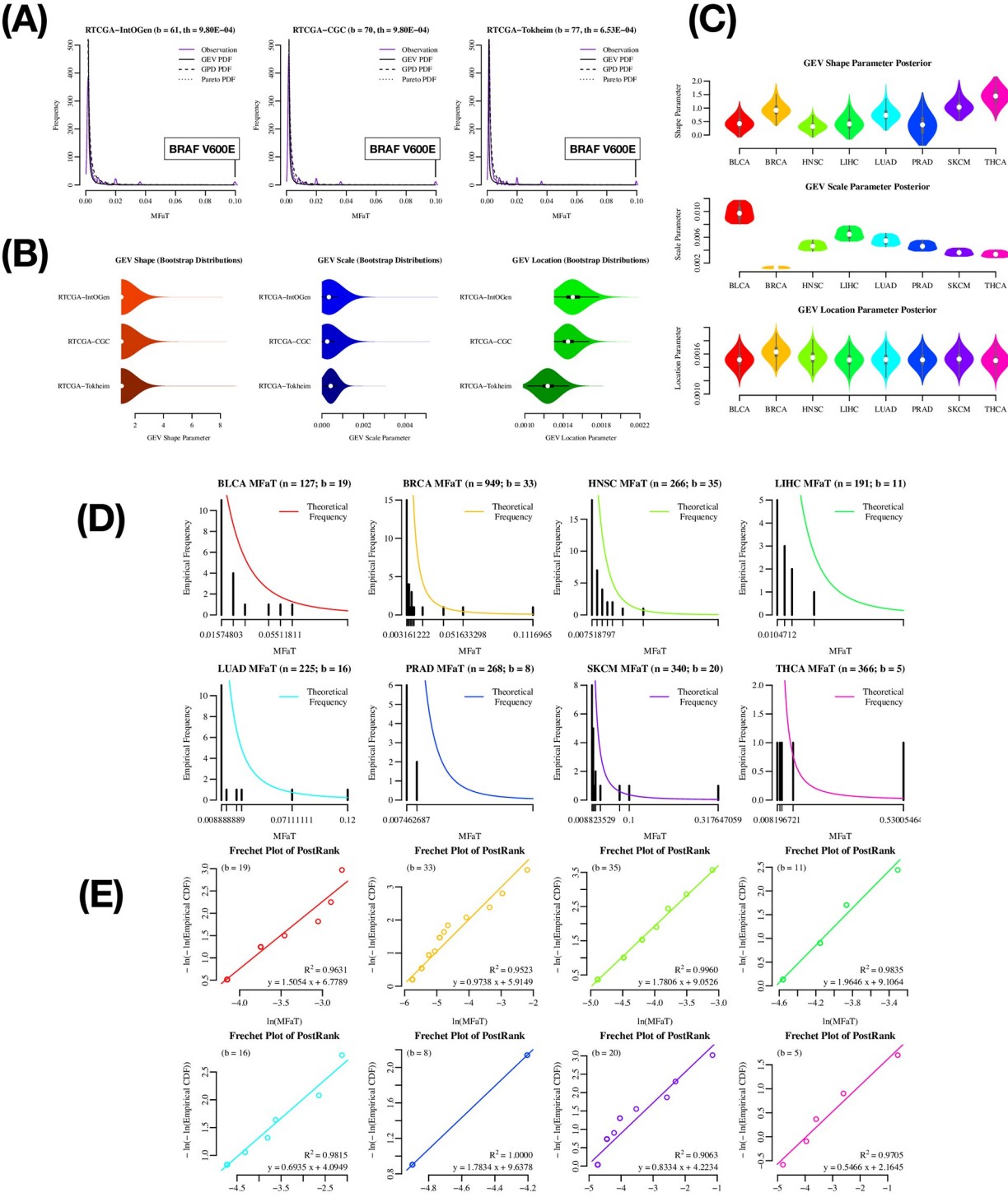

**Fig 3. Bayesian MCMC approach for estimating tumor-type-specific GEV parameters.** In these analyses, only mutations filtered by gene symbols are analyzed (i.e., symbol-based filtering). In the total-tumor analysis, we used the IntOgen, CGC, and Tokheim driver-gene definitions, and in the case of tumor-type-specific analyses, we used only the IntOgen definition. (A) Density plot of MFaTs in the cases of the RTCGA total-tumor with symbol-based filtering. The colored solid line is the probability density of observations, the black solid line is the probability density function (PDF) of GEV, and the dotted lines are the PDFs of the GPD and Pareto distributions, respectively. For each plot, a gene symbol and an amino acid substitution of a mutation with the highest MFaT value is shown in a box. Here, "b" denotes the number of genomic sites of beneficial mutations considered, and "th" denotes the effective threshold against MFaTs when selecting mutations according to ranks (for details, see Materials and methods). (B) Bootstrap distributions of GEV parameters in the total-tumor analysis. We used the maximum likelihood method for

the parameter estimation. The white point indicates the median of the distribution. The black square shows the first and the third quantiles. (C) Bootstrap distributions of GEV parameters in the tumor-type-specific analysis. We used the Bayesian MCMC method for the parameter estimation. The white point indicates the median of the distribution. The gray square shows the first and the third quantiles. (D) MFaT histograms with tumor-type-specific GEV densities. Black bars show frequencies in the histogram. The colored lines are estimated densities. (E) Type-specific Fréchet plots. The $R^2$ values and the equations of the linear regression lines are shown. Here, "b" denotes the number of genomic sites of beneficial mutations considered.

points in Fréchet plots are approximately linear ($R^2 > 0.90$ in any of eight tumor types), suggesting that driver MFaTs follow the Fréchet distribution (Fig 3E). Finally, the results of the $\chi^2$ goodness-of-fit test did not reject the null hypothesis that the distribution is GEV, except for the case of THCA (Table 3).

Collectively, the parameter estimation by the Bayesian MCMC approach confirmed that the probability distributions of tumor-type-specific driver MFaTs are also of the Fréchet type, as shown for the total tumors (Figs 2 and 3A and 3B, S2 Fig).

## GEV-binomial model estimates mutational selection coefficients (MSCs) of cancer driver protein mutations

Here, we assessed, through the estimation of MSCs, the degree of contribution of each amino acid mutation in driver proteins to oncogenesis by applying the Bayesian approach in the framework of EVT using the RTCGA mutations dataset. We recognize that a set of assumptions are required for estimating MSCs through MFaTs in the framework of EVT. The details of such assumptions are provided in the Discussion section.

In this series of analyses, we were able to compare these mutational fitness effects estimated by mutation frequencies between tumor types, because the estimators utilize MFaTs normalized by respective sample counts. This normalization thus far is independent of the classes of genes and other features of interest. Thus, EVT as a field of population genetics is consistent with the quantitative comparison of mutational fitness effects among tumor types involving both OGs and TSGs.

For example, we estimated the MSCs of *BRAF* mutations among various tumor types (Fig 4A). The estimated MSC was the highest for *BRAF* V600E in thyroid cancer (THCA), followed by *BRAF* V600E in skin cutaneous melanoma (SKCM). The *BRAF* V600E mutation in lung adenocarcinoma (LUAD) or other amino acid changes in *BRAF* showed relatively low MSCs.

**Table 3. The $\chi^2$ goodness-of-fit test *p*-values of the RTCGA dataset with tumor-type-specific *symbol*-based filtering (rank $< 50$).**

| Dataset | Tumor Type | GEV *p*-value |
|---------|-----------|---------------|
| RTCGA | BLCA | 1.000 |
| RTCGA | BRCA | 0.434 |
| RTCGA | HNSC | 1.000 |
| RTCGA | LIHC | 1.000 |
| RTCGA | LUAD | 0.506 |
| RTCGA | PRAD | 1.000 |
| RTCGA | SKCM | 0.381 |
| RTCGA | THCA | 0.002 |

Note that in this setting, a higher *p*-value such as $p = 0.50$ weakly supports a notion that the observed and theoretical distributions are alike. In this table, *p*-values are above 0.30 except for the case of THCA.

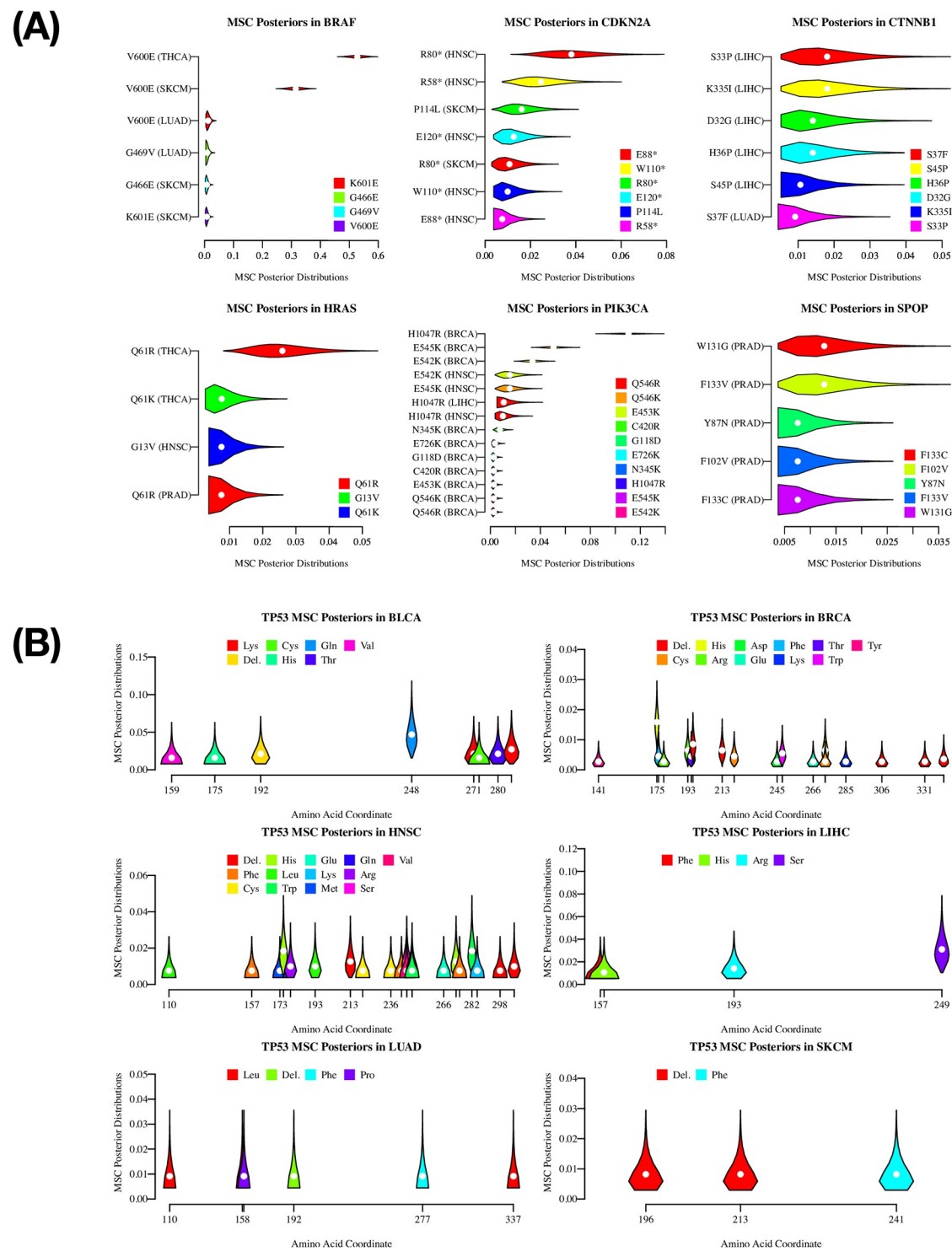

**Fig 4. The EAP estimates of driver mutation MSCs.** Tumor-type-specific posterior distributions of mutational selection coefficients (MSCs) and EAP (expected à posteriori) estimates in driver-protein mutations calculated by the GEV-binomial model (for details, see Materials and methods). For ease of plotting, we discretized probability densities of posteriors to obtain probability masses. Among the probability masses that count the total number of tumor samples (i.e., the parameter *m* in the GEV-binomial model), we omitted those tail probabilities that are smaller than 1/1000 in the plot. The white points at the center of each violin in the plots represent EAP estimates given the distribution. The shapes of plotted distributions have information of MFaT tails that cannot be modeled by a simple binomial model. (A) Violin plots of tumor-type-specific posterior distributions of MSCs in genes. Violins are colored according to each amino acid substitution to enable a comparison among MSC distributions of identically coordinated

mutations observed in different tumor types. Genes that have four "mutation-tumor type" combinations or more in the RTCGA dataset are shown. (B) Violin plots of tumor-type-specific posterior distributions of MSCs in the *TP53* protein sequences. The violins are arranged according to the amino acid coordinates of the mutations and are colored according to the amino acid residue after the substitution introduced by the mutation.

In the literature, the impact on the fitness effect of the *BRAF* V600E mutation is likely different among tumor types, including thyroid cancer, skin cutaneous melanoma, and lung adenocarcinoma. The mutation has been associated with poor prognosis and mortality in patients with papillary thyroid cancer [87–89]. This is likely the strongest association between these three tumor types. The prevalence of this mutation is reported also in melanoma [90] and is shown to induce metastasis of melanoma in mice [91]. However, this is conditional to *PTEN* loss, suggesting weaker association compared to the case of thyroid cancer. The contribution of this mutation is even weaker in lung adenocarcinoma [92]. The estimated MSCs of *BRAF* mutations among tumor types are consistent with these known facts.

Although not much is known for the impact of each mutations on the fitness effect among the tumor types for other genes in Fig 4A, OGs (i.e., *HRAS* and *PIK3CA*) tend to have a few "mutation-tumor type" combinations that show relatively high MSCs. In contrast, in the case of TSGs (i.e., *CDKN2A*, *CTNNB1*, and *SPOP*), MSCs were small and the differences between "mutation-tumor type" combinations are also small. This tendency is evident in the case of *TP53* (Fig 4B).

## Discussion

### Driver MFaTs are expectations of VAFs in the Big Bang model of cancer evolution

According to the Big Bang model of the cancer genome [2], individual tumor samples have independent and different oncogenic trajectories. Moreover, a driver mutation that has had an impact on carcinogenesis within a single sample is a "public" mutation shared by all cells in the tumor. In one sample, a class of driver mutations may have an impact on carcinogenesis, while in another sample, it does not. If a mutation at a certain driver-site within a tumor sample has an impact on carcinogenesis, then all of the tumor cells have that mutation, and about that driver site, VAF = 1.0 holds if the tumor sample purity is ideal (i.e., 100%). Similarly, if a mutation at a certain driver site within a tumor sample did not impact the oncogenic process, that mutation should not be observed in any of the tumor cells, and about that driver site, VAF = 0.0 holds regardless of tumor sample purity.

If these two important aspects of cancer driver mutation VAFs are considered, the value of aggregated tumor VAF (i.e., VAF that is aggregated across tumor samples in a given dataset; for details, see Supplementary Materials in S1 Appendix) will be equal to the value of mutant allele frequency among tumors (MFaT), which is given by the ratio of mutated samples to total samples. The use of MFaT will thus be a powerful approach in normalizing, investigating, and deciphering the records of preclonal evolution in large-scale cancer genome data.

### The ten assumptions in extreme value cancer genomics

With the framework of SSWM (strong selection and weak mutation) in population genetics [5, 25], we were able to mechanistically and stochastically describe the preclonal evolution of cancer. To achieve this and perform valid extreme value analysis over cancer driver MFaTs, we propose the following ten assumptions. Some of the relationships among these assumptions

are shown in Fig 1D. These assumptions will specify the scope of the application of the theory and will enable precise interpretation of the results.

1. **The Single Macro-Environment Assumption**: Cancer evolution is an evolutionary process in which cancer evolves and adapts to a single macro-environment. This is based on a notion that, from a viewpoint that considers multiple sets of patients and multiple tissue types, the oncogenic and progressive processes are repeatable (the macro-repeatability assumption). This also is a prerequisite for two other assumptions (i.e., the *proportionality assumption* and the *maximization assumption*) stated below.

2. **The Infinite Micro-Environments Assumption**: In a microscopic point of view that focuses on patients' genetic background, physical condition, tissue type, and tissue micro-structure, as well as the genetic diversity of cancer itself, the uncertainty of evolutionary processes, and many other critical aspects of cancer evolution, we have infinite cases of possible cancer micro-environments. This is a prerequisite of the independent and identical distribution assumption and the *maximization assumption* stated below.

3. **The Additive Dominance Assumption**: To translate knowledge of population genetics of monoploid organisms to cancer evolution in which diploid cancer cells play major roles, we assume that, in preclonal evolution of cancer cells, the fitness effect of a heterozygous allele is approximately additive [93] to explain the fitness effect of the two associated homozygotes [31]. This approximation is useful when fitness effects of beneficial mutations are small [31, 35, 36] like those mutations before violation of the *weak mutation assumption*. For activating mutation alleles in OGs, we assume that as long as there are only wildtype or heterozygous alleles in the population, effects of these alleles are additive considering the effect of wildtype as zero. Activating mutations in OGs may violate the *weak mutation assumption* while allowing us to retain the *additive dominance assumption*. For loss-of-function mutation alleles in TSGs, violation of the *additive dominance assumption* is equivalent to mutation homozygosity [55] or LoH [48, 56, 57] as a carcinogenic event.

4. **The No Epistasis Assumption**: Similar to the *additive dominance assumption*, we assume that mutation effects at different genomic loci exhibit no epistasis and thus are additive [30]. Although the framework itself is capable of incorporating epistasis by reassigning selection coefficients at each evolutionary step [30, 94], we focus on a fact that the *no epistasis assumption* is practically not overly violated, possibly because the majority of the tumor samples has driver mutations on non-redundant pathways. It is known that for tumorigenesis, at least six cancer hallmarks should be established with corresponding pathways affected either by driver mutations or by other tumorigenic events [95].

5. **The Strong Selection Assumption**: The mutations in the scope are all either beneficial or deleterious, and no neutral mutations are considered [5]. Because we presume driver mutations are all beneficial in cancer evolution [18], we can safely accept this assumption over driver mutations. In the analysis, the validity of this assumption is ensured by removing mutations with lower observed fitness gains. For the case of total-tumor analysis with symbol-based filtering, we focused on mutation sites with the top 200 MFaT values, and for type-specific analysis, we focused on the top 50.

6. **The Weak Mutation Assumption**: The beneficial mutations are fixed independently in the population [25]. This is equivalent to the exclusion of clonal interference from the scope. In the stage of preclonal evolution of cancer, it is plausible to assume that the probability of acquiring multiple mutations within a single generation is sufficiently low. This leads to an approximation of the presence of a single clone at any time. In combination with the *strong*

*selection assumption*, we obtain the SSWM fitness landscape in which the population adapts to the environment following a series of selective sweep.

7. **The Proportionality Assumption**: From the above-stated *single macro-environment assumption*, cancer adapts and evolves in a single macro-environment. Here, we assume that MFaT is proportional to the fitness gain that a mutation yields in the single macro-environment. This enables quantification of mutational fitness gains in the cancer macro-environment using MFaT.

8. **The Continuous Fitness Effect Assumption (Continuity Assumption)**: So far, the fitness effect that a mutation yields in cancer evolution is only indirectly observable. Thus, the precise formulation of the probability distribution of that variable is unknown as well as its existence. However, the repeatability of cancer evolution implies that such fitness effects by a mutation have a certain probability distribution, and the complexity suggests that the variable is approximately continuous. From the above, we assume that the mutational fitness gains in cancer evolution have a certain continuous probability distribution.

9. **The Independent and Identical Distribution Assumption (IID Assumption)**: In general, if two mutations had different genomic coordinates, then the phenotypic effects of the two mutations also vary. This is because different genomic sites encode different structures and functions of the organism. For example, mutations in the first and third letters of triplets in the codon table will yield different amino acid substitutions (i.e., the first- and third- letter substitutions are *independent*). In contrast, the fitness effect that a phenotypic effect of a given mutation confers on the organism is dependent on the environment in which the organism adapts and evolves. From the above-mentioned *infinite micro-environments assumption*, we have numerous cases of such environments in cancer evolution. Under these possible environments, we assume that the fitness effect of a given, single mutation has a certain probability distribution that is independent of a genomic site of the mutation (i.e., any given mutation have *identical* probability distribution). Then, the value of the fitness effect of a mutation is independent and identically distributed (i.i.d.) across genomic sites. This is equivalent to excluding cases of interaction of mutation effects (i.e., epistasis) from the scope.

10. **The Maximization Assumption**: From the *infinite micro-environments assumption*, the number of cases of possible cancer micro-environments is infinite. Fitness effect caused by a mutation at a given genomic site have a different value in a different micro-environment. It is known that, in the preclonal evolution of cancer, such micro-environments play critical roles in the evolution of cancer cells. Here, from the *single macro-environment assumption*, we consider adaptation of cancer cells to the single cancer macro-environment in the preclonal evolution step. We consider that, under such a macro-environment, cancer cells are selected based on combinations of different micro-environments and different fitness effects of cancer driver mutations. Thus, we assume that mutational fitness gain at a given cancer driver site is maximized across possible values (the block maxima model) after selection in the preclonal evolution. This is consistent with the idea of "survival of the fittest" in the theory of natural selection.

$$S_i = \max(S_{i,1}, S_{i,2}, \ldots, S_{i,n}) \qquad (n \to \infty) \tag{21}$$

Here, *S* is a selection coefficient, *i* is an index for genomic sites, and *n* is the number of possible micro-environments.

Although some of the above assumptions may not fit with our current knowledge of cancer biology, the results of our analysis suggest that they may hold at least for the first approximation.

## The probability distribution of cancer driver MFaTs is likely the Fréchet type

At the current level of our knowledge, statistical properties of fitness effect distributions of cancer driver mutations are to a large extent unknown. A primary reason for this is the lack of a method that directly measures such fitness effects. However, the population genetics of cancer cells based on SSWM and extreme value theory has the flexibility to deal with the behavior of cancer cell populations while avoiding this problem. With these frameworks and several appropriate assumptions, we are able to discuss at least some of the properties of the fitness effect distributions from observed mutation frequencies as a result of adaptation and evolution of cells, without knowing the exact fitness effect distribution in each mutation in each cell. In other words, if the distribution of interest is continuous, the distribution of maxima drawn from samples from that distribution will be one of these three types: Gumbel, Fréchet, or Weibull [96].

Many "ordinary" probability distributions, such as normal, exponential, and gamma, belong to the Gumbel maximum domain of attraction. Based on this fact and discussion that Fréchet-type and Weibull-type distributions are not "biological," Gumbel-type distributions have been justified as distributions of fitness effects of beneficial mutations [97]. In addition, a historical background in which such a fitness effect distribution has been considered to be exponential also supported this preconception (Fisher's geometric model; [27, 98]). However, recent theoretical advances clarified that distributions that belong to the Fréchet and Weibull domains are also possible [73].

Biological experiments roughly supported this non-Gumbel hypothesis of fitness effects of beneficial mutations. A recent experiment involving two virus strains showed that fitness effects practically yielded by a beneficial mutation do not follow an exponential distribution [99–101]. The mathematical background of this experiment is that, if values of the fitness effects have a right-truncated distribution due to their upper limit being characteristic of a given experimental setting, then the maxima of values drawn from that distribution will follow the Weibull distribution. In this experiment, a fitness effect value of a mutation is quantified as a count of formed plaques and is a proxy for virus particles [102, 103]. A quantity based on a count of biological entities is among the most powerful candidates for a variable to quantify the fitness effects of a mutation.

Also, an *Escherichia coli* experiment designed as an application of EVT empirically confirmed that the fitness effects of fixed beneficial mutations follow a distribution with a positive mode [104]. Although experimental settings including the method to quantify mutant fitness are greatly different from this study, the Fréchet distribution as a statistical distribution that describes the behavior of fitness effects of fixed driver mutations in tumor samples also has a positive mode in its mathematics.

Our study suggested that the distribution of fitness effects of driver mutations calculated from a sample frequency in a large-scale sequence dataset is of the Fréchet type (Figs 2 and 3), while it also allows distributions of the fitness effects of the individual mutations to remain unknown. The results of goodness-of-fit tests (Tables 1–3) did not reject the null hypothesis that the given two distributions are identical, supporting the possibility that the distribution of MFaTs as estimates of mutational selection coefficients (MSCs) is Fréchet distribution. Mathematically, a zero value of the shape parameter of the generalized extreme value (GEV)

distribution means the Gumbel type, and a positive value means the Fréchet type. Also, the Fréchet distribution itself belongs to the attraction domain of the Fréchet distribution. These results not only present a problem to the previously held Gumbel hypothesis [73] from a practical point of view but also suggest the applicability of the Fréchet distribution in cancer genomics (Fig 4). In the case of THCA (Table 3), the null hypothesis was rejected in the goodness-of-fit test and it did not reproduce this result. It is obvious from the graph that this irreproducibility is due to the lack of mutations used in the analysis (Fig 3E).

## The applicability of extreme value theory in cancer genomics

The posterior distributions of tumor-type-specific mutational selection coefficients (MSCs) of driver mutations calculated by the GEV-binomial model (Fig 4) contain information of distribution tails described by EVT. In the violin plots, because the EAP estimates drawn as white dots contain information of the tails that cannot be handled by a simple binomial model, those estimates have shifted to the right from the central point, as suggested by shapes of the posterior distributions. Such shifts are significant in posterior distributions of mutations, such as the S33P mutation in the *CTNNB1* gene in the LIHC tumor type and the Q61R mutation in the *HRAS* gene in the THCA tumor type (Fig 4A). Similarly, while EAP estimates of driver mutation MSCs in the *TP53* gene strongly reflect MFaTs that are mutant sample frequencies, these values also reflect information of the tails so as to be more continuous (Fig 4B). The shifts in these estimates suggest the applicability of EVT in cancer genomic analyses that entail estimation of fitness effects of beneficial mutations.

## Conclusion

Based on statistical data analysis involving multiple tumor types and multiple definitions of cancer driver mutations, this study not only demonstrates that EVT helps us to understand the statistical distribution of driver-mutation frequencies in the cancer genome, which is a critical aspect in cancer genetics, but also suggests its applicability in cancer genomics based on its potential to model the tail behavior of mutation frequency distributions.

## Supporting information

**S1 Fig. The SSWM population dynamics of cells and the additivity of fitness effects of mutant alleles.**
(PDF)

**S2 Fig. Exploratory plots on cancer driver mutation MFaTs in the total-tumor analysis with position-based filtering.**
(PDF)

**S1 Appendix. Supplementary materials.**
(PDF)

**S1 Script. Get_IcgcProteinAltering.py.**
(PY)

**S1 File. Database_ICGC_temp_PostMax.tsv.gz.**
(GZ)

**S2 File. Database_COSMIC_temp_PostMax.tsv.gz.**
(GZ)

**S3 File. Database_CHANG_temp_PostMax.tsv.gz.**
(GZ)

**S4 File. Driver_TotalGene_database_IntOGen.tsv.**
(TSV)

**S5 File. Driver_TotalGene_database_CGC.tsv.**
(TSV)

**S6 File. Driver_TotalGene_database_Tokheim.tsv.**
(TSV)

**S7 File. Driver_TotalSite_database_IntOGen.tsv.**
(TSV)

**S8 File. Driver_TotalSite_database_DoCM.tsv.**
(TSV)

**S9 File. Driver_TotalSite_database_Bailey.tsv.**
(TSV)

**S10 File. Database_RTCGA_temp_Format.tsv.gz.**
(GZ)

**S11 File. Database_RTCGA_temp_ProteinAlteringSnv.tsv.gz.**
(GZ)

**S12 File. Database_RTCGA_temp_ProteinAlteringSnvDoubleton.tsv.gz.**
(GZ)

**S13 File. Database_RTCGA_temp_PostMaxTotal.tsv.**
(TSV)

**S14 File. Database_RTCGA_temp_PostMaxType.tsv.**
(TSV)

**S15 File. Driver_TypeGene_database_IntOGen.tsv.**
(TSV)

**S16 File. Drivertype_RTCGA-IntOGen_temp_PreRank.tsv.**
(TSV)

**S17 File. R_analysis_TotalTumor.zip.**
(ZIP)

**S18 File. R_analysis_TumorTypeSpecific.zip.**
(ZIP)

**S19 File. Table_NormalizedEap.tsv.**
(TSV)

## Acknowledgments

NT thanks Takao Suetake for his support and encouragement throughout this research.

## Author Contributions

**Conceptualization:** Natsuki Tokutomi.

**Formal analysis:** Natsuki Tokutomi.

**Funding acquisition:** Sumio Sugano.

**Investigation:** Natsuki Tokutomi.

**Methodology:** Natsuki Tokutomi.

**Project administration:** Kenta Nakai, Sumio Sugano.

**Resources:** Natsuki Tokutomi.

**Software:** Natsuki Tokutomi.

**Supervision:** Kenta Nakai, Sumio Sugano.

**Validation:** Natsuki Tokutomi.

**Visualization:** Natsuki Tokutomi.

**Writing – original draft:** Natsuki Tokutomi.

**Writing – review & editing:** Sumio Sugano.

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
