## [Decision Letter · Decision Letter 0]

23 Dec 2020

PONE-D-20-36757

Extreme value theory as a general framework for understanding mutation frequency distribution in cancer genomes

PLOS ONE

Dear Dr. TOKUTOMI,

Thank you for submitting your manuscript to PLOS ONE. After careful consideration, we feel that it has merit but does not fully meet PLOS ONE’s publication criteria as it currently stands. Therefore, we invite you to submit a revised version of the manuscript that addresses the points raised during the review process.

Please check the reviewers' comments and address all the issues. Otherwise, the manuscript will not be considered for publication. The main issues include the readability of the manuscript, lack of referring to current research, and insufficient biological insights besides other issues. 

We look forward to receiving your revised manuscript.

Kind regards,

Zechen Chong

Academic Editor

PLOS ONE

Journal Requirements:

Reviewers' comments:

Reviewer's Responses to Questions

**Comments to the Author**

1. Is the manuscript technically sound, and do the data support the conclusions?

Reviewer #1: Yes

Reviewer #2: Partly

2. Has the statistical analysis been performed appropriately and rigorously? 

Reviewer #1: Yes

Reviewer #2: I Don't Know

3. Have the authors made all data underlying the findings in their manuscript fully available?

Reviewer #1: Yes

Reviewer #2: Yes

4. Is the manuscript presented in an intelligible fashion and written in standard English?

Reviewer #1: Yes

Reviewer #2: No

5. Review Comments to the Author

Reviewer #1: The authors present a framework based on extreme value theory to model cancer mutation frequencies. They demonstrate that this model explains the distributions observed in several large cancer genomic datasets.

1. The main issue is the insufficient framing of the study within existing work, beginning with the abstract claiming that "Currently, there is no recognized population genetics framework describing the population dynamics of cancer cells that is applicable to real cancer genome data." The meaning of the last part of the sentence is unclear to me, but in any case, there have been many studies analyzing cancer mutations under population genetics frameworks. For example:

Iwasa, Y., Michor, F., Komarova, N.L. and Nowak, M.A., 2005. Population genetics of tumor suppressor genes. Journal of theoretical biology, 233(1), pp.15-23.

Attolini, C.S.O., Cheng, Y.K., Beroukhim, R., Getz, G., Abdel-Wahab, O., Levine, R.L., Mellinghoff, I.K. and Michor, F., 2010. A mathematical framework to determine the temporal sequence of somatic genetic events in cancer. Proceedings of the National Academy of Sciences, 107(41), pp.17604-17609.

Durrett, R. Population genetics of neutral mutations in exponentially growing cancer cell populations. Ann. Appl. Probabil. 23, 230–250 (2013).

Hu, Z., Sun, R. and Curtis, C., 2017. A population genetics perspective on the determinants of intra-tumor heterogeneity. Biochimica et Biophysica Acta (BBA)-Reviews on Cancer, 1867(2), pp.109-126.

Niida, A., Iwasaki, W.M. and Innan, H., 2018. Neutral theory in cancer cell population genetics. Molecular biology and evolution, 35(6), pp.1316-1321.

Caravagna, G., Heide, T., Williams, M. J., Zapata, L., Nichol, D., Chkhaidze, K., ... & Chesler, L. (2020). Subclonal reconstruction of tumors by using machine learning and population genetics. Nature Genetics, 52(9), 898-907.

The Introduction cites studies in reference to elements of the proposed framework, but omits much research that seems similar to the study as a whole. And in regards to extreme value theory, a paper with important background on EVT applied to beneficial mutations is listed in the references (#41) but doesn't seem to actually be cited in the manuscript.

Without such connections to related existing research, it is unclear whether the framework is novel or replicates existing methods.

2. MFaT is defined in two subsections ("The Definition and Calculation of MFaT" in Methods and "The Definition of MFaT" in Results). This seems excessive, especially since it gives the impression of a novel metric, even though the fraction of tumors with a particular mutation is a very common statistic in cancer genomics studies.

3. Page 17: "In the field of population genetics, a lot of effort has been devoted solely to monoploid organisms. To translate this knowledge to population genetics of cancer cells, we need an additional assumption for the fitness of heterozygotes: the fitness of a heterozygote is exactly intermediate between the two associated homozygotes." This seems to imply that diploid organisms have been insufficiently studied in population genetics, when in fact, much of population genetics has been devoted to the study of diploid organisms (especially humans), with dominance being a crucial phenomenon in the models. Similarly, the assumption of heterozygote fitness being exactly intermediate is rarely true for cancer mutations.

4. Page 17/18: The framework assumes there is no genomic epistasis, but important cancer mutations often exhibit epistasis, for example between mutations with redundant effects in the same pathway. More justification should be given that this assumption is not overly violated in practice.

Minor points:

5. Gene names should be italicized.

6. Line 614, caner -> cancer

Reviewer #2: The manuscript by Tokutomi et al uses extreme value theory (EVT) to help explain driver mutation frequencies in large public cancer datasets. The paper is interesting, but has several shortcomings. This reviewer is more qualified to comment on the more biological aspects of this manuscript.

1) The manuscript is hard to understand, especially for a more general audience of PLOS ONE, versus a more specialized journal. In particular, the “model” is not very well-described. The data appear to be mutation frequencies, where a limited number of driver mutations (in aggregate) have high mutation frequencies (MFaT) but most driver mutations have very low mutation frequencies. This phenomenon is well-described and well-known. The manuscript could better describe, perhaps with a cartoon exactly what is being modeled biologically. Otherwise, it appears to be mainly a curve-fitting type of exercise, which is interesting but of uncertain significance.

2) The data do not adjust for sample purity, which will reinforce the low frequency driver mutations

3) It seems unlikely that “all” cancers follow the same rules or even have the same driver mutations. Yet the manuscript combines multiple types of cancer. It is uncertain if the analysis works better for individual cancer type, or if the analysis works better when combining all the data (ie as in Figure 1). It would be helpful if some of the peaks with high MFat (TP53?) could be identified in Fig1 A.

4) Along these lines, it is unclear if different mutations in the same gene but at different nucleotides are counted as a “driver” mutation. (Does TP53 count as a single driver or can it be different drivers?---this reviewer has trouble with the description on page 4)

5) It would be interesting or perhaps informative if the authors also looked at passenger mutations (such as TTN) to check if the distributions do or do not follow an EVT type of distribution. This is optional, but could be interesting because of the uncertainty of whether many low frequency “driver” mutations are in fact driver mutations.

6) The paper concludes that EVT and the analysis can help estimate the fitness of beneficial mutations. It would be useful if the authors provide a list of beneficial mutations and their estimated fitness from their analysis (if possible).

6. PLOS authors have the option to publish the peer review history of their article (what does this mean?). If published, this will include your full peer review and any attached files.

Reviewer #1: No

Reviewer #2: No

---

## [Author Response · Author response to Decision Letter 0]

8 Mar 2021

First, we wish to thank reviewers for providing critical insights and many good suggestions. We revised our manuscript based on these suggestions. We think our manuscript has been improved in terms of structure, clarity, correctness, and connection to existing research.

The point-to-point response:

Reviewer #1:

Point #1-1: Insufficient framing and connection.

The main issue is the insufficient framing of the study within existing work, beginning with the abstract claiming that "Currently, there is no recognized population genetics framework describing the population dynamics of cancer cells that is applicable to real cancer genome data." The meaning of the last part of the sentence is unclear to me, but in any case, there have been many studies analyzing cancer mutations under population genetics frameworks. For example:

Iwasa, Y., Michor, F., Komarova, N.L. and Nowak, M.A., 2005. Population genetics of tumor suppressor genes. Journal of theoretical biology, 233(1), pp.15-23.

Attolini, C.S.O., Cheng, Y.K., Beroukhim, R., Getz, G., Abdel-Wahab, O., Levine, R.L., Mellinghoff, I.K. and Michor, F., 2010. A mathematical framework to determine the temporal sequence of somatic genetic events in cancer. Proceedings of the National Academy of Sciences, 107(41), pp.17604-17609.

Durrett, R. Population genetics of neutral mutations in exponentially growing cancer cell populations. Ann. Appl. Probabil. 23, 230–250 (2013).

Hu, Z., Sun, R. and Curtis, C., 2017. A population genetics perspective on the determinants of intra-tumor heterogeneity. Biochimica et Biophysica Acta (BBA)-Reviews on Cancer, 1867(2), pp.109-126.

Niida, A., Iwasaki, W.M. and Innan, H., 2018. Neutral theory in cancer cell population genetics. Molecular biology and evolution, 35(6), pp.1316-1321.

Caravagna, G., Heide, T., Williams, M. J., Zapata, L., Nichol, D., Chkhaidze, K., ... & Chesler, L. (2020). Subclonal reconstruction of tumors by using machine learning and population genetics. Nature Genetics, 52(9), 898-907.

The Introduction cites studies in reference to elements of the proposed framework, but omits much research that seems similar to the study as a whole. And in regards to extreme value theory, a paper with important background on EVT applied to beneficial mutations is listed in the references (#41) but doesn't seem to actually be cited in the manuscript.

Without such connections to related existing research, it is unclear whether the framework is novel or replicates existing methods.

Response to the Point #1-1:

We rewrote and re-organized the Introduction section in order to clarify the framing of our work within antecedent studies.

More than 50 new reference papers including six papers that the reviewer kindly referred to are cited in the Introduction section.

The EVT background paper (formerly reference #41, in this revised manuscript #20, Orr HA, 2010) is cited twice (L29 and L106).

In addition, we included the following elements in the Introduction to provide sufficient connection to existing research.

An important population genetics model (the mutation-selection balance; MSB) that seems similar to the strong selection and weak mutation (SSWM) model in context of cancer evolution (L23-L44).

SSWM model applications to cancer evolution after the clonal expansion (L51-L63).

Knudson's Two Hit hypothesis (L19-L22).

Mode of somatic cell division (L139-L151).

Point #1-2: MFaT is doubly described.

MFaT is defined in two subsections ("The Definition and Calculation of MFaT" in Methods and "The Definition of MFaT" in Results). This seems excessive, especially since it gives the impression of a novel metric, even though the fraction of tumors with a particular mutation is a very common statistic in cancer genomics studies.

Response to the Point #1-2:

We removed one MFaT definition from the Results section. The related description is merged to the MFaT definition in the Materials and Methods section (L312-L335). Although the fraction of tumors with a particular mutation is a common statistic, we need to avoid confusion with another common statistic, VAF. To achieve this, explanatory paragraphs are added to the Introduction section (L232-L247). 

Point #1-3: Limited description of diploid genetics.

Page 17: "In the field of population genetics, a lot of effort has been devoted solely to monoploid organisms. To translate this knowledge to population genetics of cancer cells, we need an additional assumption for the fitness of heterozygotes: the fitness of a heterozygote is exactly intermediate between the two associated homozygotes." This seems to imply that diploid organisms have been insufficiently studied in population genetics, when in fact, much of population genetics has been devoted to the study of diploid organisms (especially humans), with dominance being a crucial phenomenon in the models. Similarly, the assumption of heterozygote fitness being exactly intermediate is rarely true for cancer mutations.

Response to the Point #1-3:

We clarified the related assumption, approximation and condition. The item "The Additivity Assumption" is renamed to "The Additive Dominance Assumption" to clarify the extent to which the assumption remains approximately valid (L653-L666). Mutations in oncogenes and tumor suppressor genes are treated as exceptions (L660-L666). In addition, the dominance coefficient in relation to cancer biology is discussed in the Introduction section (L29, L42, and L97-L151). 

Point #1-4: Lack of justification on no genomic epistasis.

Page 17/18: The framework assumes there is no genomic epistasis, but important cancer mutations often exhibit epistasis, for example between mutations with redundant effects in the same pathway. More justification should be given that this assumption is not overly violated in practice.

Response to the Point #1-4: An item is added.

A new item "The No Epistasis Assumption" is added to the Discussion section (L667-L675). In the item, the complexity of tumor phenotype and its associated pathways are discussed (L670-L675).

Point #1-5: Italicizing gene symbols

Gene names should be italicized.

Response to the Point #1-5:

We italicized gene names.

Point #1-6: Spell missing (caner -> cancer)

Line 614, caner -> cancer

Response to the Point #1-6:

We corrected the misspelling.

Reviewer #2:

Point #2-1: Cartoon of the model.

The manuscript is hard to understand, especially for a more general audience of PLOS ONE, versus a more specialized journal. In particular, the “model” is not very well-described. The data appear to be mutation frequencies, where a limited number of driver mutations (in aggregate) have high mutation frequencies (MFaT) but most driver mutations have very low mutation frequencies. This phenomenon is well-described and well-known. The manuscript could better describe, perhaps with a cartoon exactly what is being modeled biologically. Otherwise, it appears to be mainly a curve-fitting type of exercise, which is interesting but of uncertain significance.

Response to the Point #2-1: A new figure is added.

A new figure with four panels including cartoons is added so as to clarify our model (Figure 1). And this figure was referred to several times in the Introduction (L67, L102-L104, and L130) section. At the same time, we reorganized and rewrote the Introduction section so as to clarify our assumptions for the models as well as the model itself. More than 50 new reference papers cited in the Introduction section in order to clarify the framing of our work within antecedent studies. In addition, we included the following elements in the Introduction to provide sufficient connection to existing research.

An important population genetics model (the mutation-selection balance; MSB) that seems similar to the strong selection and weak mutation (SSWM) model in context of cancer evolution (L23-L44).

SSWM model applications to cancer evolution after the clonal expansion (L51-L63).

Knudson's Two Hit hypothesis (L19-L22).

Mode of somatic cell division (L139-L151).

Point #2-2: Lack of reference to adjustment of sample purity.

The data do not adjust for sample purity, which will reinforce the low frequency driver mutations

Response to the Point #2-2: Reference added.

As the reviewer implied, large-scale genomics datasets have different levels of sample purity. Therefore, we restricted ourselves to separated analysis within each dataset. Adjusting datasets each containing tens of thousands of specimens for sample purity is often computationally impractical, leading to a shared agreement that referring to the database name in combination with an original filtering method is sufficient. For example: 

Wong, Wing Chung, et al. "CHASM and SNVBox: toolkit for detecting biologically important single nucleotide mutations in cancer." Bioinformatics 27.15 (2011): 2147-2148.

Mao, Yong, et al. "CanDrA: cancer-specific driver missense mutation annotation with optimized features." PloS one 8.10 (2013): e77945.

Cisneros, Luis, et al. "Ancient genes establish stress-induced mutation as a hallmark of cancer." PLoS One 12.4 (2017): e0176258.

However, to ensure consistency of the sample purity, it is widely accepted to refer to the original paper of such a dataset in which mutation filtering criteria are described. Thus, we referred to such literatures in the "Data Processing" item in the Materials and Methods section (L294-L296).

Point #2-3: Unclear graph representation.

It seems unlikely that “all” cancers follow the same rules or even have the same driver mutations. Yet the manuscript combines multiple types of cancer. It is uncertain if the analysis works better for individual cancer type, or if the analysis works better when combining all the data (ie as in Figure 1). It would be helpful if some of the peaks with high MFat (TP53?) could be identified in Fig1 A.

Response to the Point #2-3:

With sincere respect to the points that the reviewer made, we improved appearance of the graphs. In all density plots of total-tumor MFaTs, mutations of the highest MFaTs are indicated (Figs 2A, 3A, and 4A).

Point #2-4: Lack of clarity and troublesome descriptions.

Along these lines, it is unclear if different mutations in the same gene but at different nucleotides are counted as a “driver” mutation. (Does TP53 count as a single driver or can it be different drivers?---this reviewer has trouble with the description on page 4)

Response to the Point #2-4:

To address this issue, we replaced old text and added a new item. The original contents of the item "Data Processing" is now moved to S1 Appendix and newer text is supplied in the item (L293-L310). In addition, a new item "Study Design" is added to clarify the method of counting driver mutations (L249-L262).

Point #2-5: Low frequency potential driver mutations.

It would be interesting or perhaps informative if the authors also looked at passenger mutations (such as TTN) to check if the distributions do or do not follow an EVT type of distribution. This is optional, but could be interesting because of the uncertainty of whether many low frequency “driver” mutations are in fact driver mutations.

Response to the Point #2-5:

The suggestion is remarkably interesting. But, in current manuscript, we wish to concentrate on establishing and presenting the EVT framework. Also, we feel one needs some careful considerations involving effect sizes of mutations (Some of our assumptions hold exclusively for driver genes). We hope to pursue this passenger mutation problem as a separate study in future. Thank you very much for your great suggestion. 

Point #2-6: The table of calculated fitness effects.

The paper concludes that EVT and the analysis can help estimate the fitness of beneficial mutations. It would be useful if the authors provide a list of beneficial mutations and their estimated fitness from their analysis (if possible).

Response to the Point #2-6:

We added tables of estimated driver fitness effects for each cancer types in S1 Appendix (Appendix Tables 1-8). In addition, the single table for these tumor types is included in the Supplementary Files (S19 File table NormalizedEap.tsv). We hope these tables will be useful.

---

## [Decision Letter · Decision Letter 1]

19 Apr 2021

PONE-D-20-36757R1

Extreme value theory as a framework for understanding mutation frequency distribution in cancer genomes

PLOS ONE

Dear Dr. Tokutomi,

Thank you for resubmitting your manuscript to PLOS ONE. The revision has been carefully reviewed by the original reviewers. However, there are still some issues have not been addressed. Therefore, we invited another reviewer to further judge. Please take seriously about the reviewers' comments and address all the issues sufficiently. 

We look forward to receiving your revised manuscript.

Kind regards,

Zechen Chong

Academic Editor

PLOS ONE

Reviewers' comments:

Reviewer's Responses to Questions

**Comments to the Author**

1. If the authors have adequately addressed your comments raised in a previous round of review and you feel that this manuscript is now acceptable for publication, you may indicate that here to bypass the “Comments to the Author” section, enter your conflict of interest statement in the “Confidential to Editor” section, and submit your "Accept" recommendation.

Reviewer #1: All comments have been addressed

Reviewer #2: All comments have been addressed

Reviewer #3: (No Response)

2. Is the manuscript technically sound, and do the data support the conclusions?

Reviewer #1: Yes

Reviewer #2: Partly

Reviewer #3: Partly

3. Has the statistical analysis been performed appropriately and rigorously? 

Reviewer #1: Yes

Reviewer #2: I Don't Know

Reviewer #3: Yes

4. Have the authors made all data underlying the findings in their manuscript fully available?

Reviewer #1: Yes

Reviewer #2: Yes

Reviewer #3: Yes

5. Is the manuscript presented in an intelligible fashion and written in standard English?

Reviewer #1: Yes

Reviewer #2: No

Reviewer #3: Yes

6. Review Comments to the Author

Reviewer #1: (No Response)

Reviewer #2: The manuscript continues to be very difficult to understand. The concepts and the logic seem insufficient for a more general readership that is seen with PLOS ONE. The added "cartoon" (Fig 1) illustrates general principles, but does not seem to allow a reader to "see" how extreme value theory is being applied in the paper.

Reviewer #3: This is an interesting paper that applies the extreme value theory to model the mutation frequency distribution in the cancer genome. The authors showed that the driver mutation count distribution can be well fitted by the Frechet-type extreme distribution across varying cancer genomic datasets (e.g. TCGA, ICGC, etc). Based on the model fitting, the authors concluded that early tumor evolution (pre-clonal expansion) likely follows the classic strong selection and weak mutation (SSWM) regime in population genetics theory.

1) It’s not clear to me regarding the relationship between SSWM model and the extreme value distribution. Can the authors provide a mathematical prove that the SSWM regime of tumor evolution will give rise to Frechet-type extreme value distribution for driver mutation number? Or others have proven this?

2) I don’t understand why the authors focused on specific site of driver genes? It seems gene-level mutation distribution fits the Frechet-type distribution better than site-level distribution (Fig 2 and 3). I suggest the authors to show gene-level mutation distribution fitting in main figures and site-level mutation distribution in supplementary data. Also, Fig 2a and 3a are both site-level fitting, why it was said Fig 2 is on gene level while Fig 3 is site level.

3) Is the dominance h (Fig. 1D) important in the mathematical analysis? I didn’t see any information regarding h in results.

4) The introduction section is too long and should be cutted and many of the information is misleading.

7. PLOS authors have the option to publish the peer review history of their article (what does this mean?). If published, this will include your full peer review and any attached files.

Reviewer #1: No

Reviewer #2: No

Reviewer #3: No

---

## [Author Response · Author response to Decision Letter 1]

22 Jul 2021

Reviewer #1: (No Response)

Reviewer #2: The manuscript continues to be very difficult to understand. The concepts and the logic seem insufficient for a more general readership that is seen with PLOS ONE. The added "cartoon" (Fig 1) illustrates general principles, but does not seem to allow a reader to "see" how extreme value theory is being applied in the paper.

We completely changed Fig 1 to clarify not only the medical and cell-biological aspects of our study, but also the point where extreme value theory (EVT) is applied to the cellular fitness landscape (Fig 1A, B, D). We also made big modification in the Introduction part. In the first section (The “Big Bang” Model of Cancer Development and Population Genetics of Cancer Cells) in Introduction, the concepts and the logic are made more straightforward for the reader's ease of understanding. We wish these changes, in combination with the schematic S1FigC in Supporting information, may allow a reader to "see" the way by which EVT is applied in the paper.

Reviewer #3: This is an interesting paper that applies the extreme value theory to model the mutation frequency distribution in the cancer genome. The authors showed that the driver mutation count distribution can be well fitted by the Frechet-type extreme distribution across varying cancer genomic datasets (e.g. TCGA, ICGC, etc). Based on the model fitting, the authors concluded that early tumor evolution (pre-clonal expansion) likely follows the classic strong selection and weak mutation (SSWM) regime in population genetics theory.

1) It’s not clear to me regarding the relationship between SSWM model and the extreme value distribution. Can the authors provide a mathematical prove that the SSWM regime of tumor evolution will give rise to Frechet-type extreme value distribution for driver mutation number? Or others have proven this?

There is no direct relationship between the SSWM model and the extreme value distribution. However, the SSWM model is important here as a prerequisite for an assumption that the adaptation process is Markov process. This allows us to assume that only mutations with fitness effects which is the maximum or above a certain threshold will fix in each adaptation step. The "above the threshold" model gives rise to Generalized Pareto Distribution (GPD); and the "maxima" model gives rise to Generalized Extreme Value distribution (GEV).

 More precisely, Joyce et al (Joyce et al, 2008) demonstrated that the fitness landscape of adaptation of DNA sequence would follow GPD under the assumption of the SSWM and the "above the threshold" model. In this paper, we used the "block maxima" model because it fitted better with the "Big Bang" model of the cancer evolution. The Fisher-Tippett-Gnedenko theorem proves that the maxima from independent and identically distributed random variables will follow GEV distributions.

(Joyce P et al, 2008): Joyce, Paul, et al. "A general extreme value theory model for the adaptation of DNA sequences under strong selection and weak mutation." Genetics 180.3 (2008): 1627-1643.

2) I don’t understand why the authors focused on specific site of driver genes? It seems gene-level mutation distribution fits the Frechet-type distribution better than site-level distribution (Fig 2 and 3). I suggest the authors to show gene-level mutation distribution fitting in main figures and site-level mutation distribution in supplementary data. Also, Fig 2a and 3a are both site-level fitting, why it was said Fig 2 is on gene level while Fig 3 is site level.

Classical population genetics theories analyze evolution of a DNA sequence as a set of nucleotide residues instead of gene symbols (e.g., (Orr HA, 2002)). To perform "gene-level" analysis, we imagine that more careful formulation and study design are necessary. Thus, in fact, all the analyses performed in this paper were at "site-level". We wish that we have a chance to design "gene-level" analysis in a separate study in future.

 We agree that the formerly Fig 3A (now S2FigA) shows lack of data points in ICGC cases. We have moved the formerly Fig 3 figures to Supporting information. Also, misleading labels in the figure and table texts (i.e., "driver-gene definition" and "driver-site definition") are now replaced with clearer ones (i.e., "symbol-based filtering" and "position-based filtering"). In addition, the corresponding descriptions in Results (L466-L467, L486-L487, L496) and Discussion (L617) are updated.

(Orr HA, 2002): Orr, H. Allen. "The population genetics of adaptation: the adaptation of DNA sequences." Evolution 56.7 (2002): 1317-1330.

3) Is the dominance h (Fig. 1D) important in the mathematical analysis? I didn’t see any information regarding h in results.

We agree that although the dominance coefficient h may serve our understanding, it is less important in the context of the Introduction part. The details of this coefficient are omitted from Introduction and its related figures are moved to Supporting information (S1FigA, S1FigB).

4) The introduction section is too long and should be cutted and many of the information is misleading.

We removed redundancy in the description and made it more straightforward, especially in the first section of the introduction. As a result, the introduction part has decreased by 24 % in word count (3088 -> 2354 words, citations and formula included). We wish the description in the section has become more concise, straightforward, and clear.

---

## [Decision Letter · Decision Letter 2]

11 Aug 2021

Extreme value theory as a framework for understanding mutation frequency distribution in cancer genomes

PONE-D-20-36757R2

Dear Dr. TOKUTOMI,

We’re pleased to inform you that your manuscript has been judged scientifically suitable for publication and will be formally accepted for publication once it meets all outstanding technical requirements.

Kind regards,

Zechen Chong

Academic Editor

PLOS ONE

Additional Editor Comments (optional):

Reviewers' comments:

Reviewer's Responses to Questions

**Comments to the Author**

1. If the authors have adequately addressed your comments raised in a previous round of review and you feel that this manuscript is now acceptable for publication, you may indicate that here to bypass the “Comments to the Author” section, enter your conflict of interest statement in the “Confidential to Editor” section, and submit your "Accept" recommendation.

Reviewer #1: All comments have been addressed

Reviewer #2: All comments have been addressed

Reviewer #3: All comments have been addressed

2. Is the manuscript technically sound, and do the data support the conclusions?

Reviewer #1: Yes

Reviewer #2: Yes

Reviewer #3: Yes

3. Has the statistical analysis been performed appropriately and rigorously? 

Reviewer #1: Yes

Reviewer #2: I Don't Know

Reviewer #3: Yes

4. Have the authors made all data underlying the findings in their manuscript fully available?

Reviewer #1: Yes

Reviewer #2: Yes

Reviewer #3: Yes

5. Is the manuscript presented in an intelligible fashion and written in standard English?

Reviewer #1: Yes

Reviewer #2: Yes

Reviewer #3: Yes

6. Review Comments to the Author

Reviewer #1: (No Response)

Reviewer #2: The authors have made a commendable effort to better explain their approach to a more general audience. The text and diagrams are much better.

Reviewer #3: My concerns have been well addressed by the authors. This is an interesting theoretical work on cancer evolution. I would like to see this manuscript available.

7. PLOS authors have the option to publish the peer review history of their article (what does this mean?). If published, this will include your full peer review and any attached files.

Reviewer #1: No

Reviewer #2: No

Reviewer #3: No

---

## [Editor Report · Acceptance letter]

13 Aug 2021

PONE-D-20-36757R2 

Extreme value theory as a framework for understanding mutation frequency distribution in cancer genomes  

Dear Dr. Tokutomi:

I'm pleased to inform you that your manuscript has been deemed suitable for publication in PLOS ONE. Congratulations! Your manuscript is now with our production department. 

Kind regards, 

on behalf of

Dr. Zechen Chong 

Academic Editor

PLOS ONE